

# Near-extremal limits of warped black holes

Ankit Aggarwal[1,2⋆], Alejandra Castro[3†], Stéphane Detournay[1‡] and Beatrix Mühlmann[4∘]

**1** Physique Mathématique des Interactions Fondamentales, Université Libre de Bruxelles and International Solvay Institutes, Campus Plaine - CP 231, 1050 Bruxelles, Belgium

**2** Institute for Theoretical Physics Amsterdam and Delta Institute for Theoretical Physics, University of Amsterdam, Science Park 904, 1098 XH Amsterdam, The Netherlands

**3** Department of Applied Mathematics and Theoretical Physics, University of Cambridge, Cambridge CB3 0WA, United Kingdom

**4** Department of Physics, McGill University, Montreal QC H3A 2T8, Canada

⋆ ankit.aggarwal@ulb.be , † ac2553@cam.ac.uk ,
‡ sdetourn@ulb.ac.be , ∘ beatrix.muehlmann@mcgill.ca

## Abstract

A holographic description of three-dimensional warped black holes suffers from ambiguities due to a seemingly harmless choice of coordinate system. This gives rise to the notion of ensembles in warped black holes, and we focus on two of them: the canonical and quadratic ensemble. Our aim is to quantify the imprint of these ensembles in the near-extremal limit of a warped black hole. To this end, for each ensemble, we explore the thermodynamic response and evaluate greybody factors. We also set-up a holographic dictionary in their near-AdS$_2$ region, and decode aspects of the dual near-CFT$_1$. This gives us different perspectives of the black hole that we can contrast and compare. On the one hand, we find perfect agreement between the near-extremal limit of the canonical ensemble warped black holes, their near-AdS$_2$ effective analysis, and a warped conformal field theory description. On the other, we are led to rule out the quadratic ensemble due to inconsistencies at the quantum level with the near-AdS$_2$ effective description.



# 1 Introduction

Warped black holes are three-dimensional stationary spacetimes, which can carry mass and angular momentum. They usually appear as classical solutions of gravitational theories with a massive degree of freedom, such as topologically massive gravity [1,2], theories with a massive vector field [3,4], or higher-derivative theories [5–7]. In all of these cases, the term "warped" originates from approximate symmetries of the solution: in the absence of the black hole, the Killing vectors of a warped background form an $sl(2) \times u(1)$ algebra. Intuitively, this algebra can be understood as a deformation, or warping, of the size of a circle fiber inside of three-dimensional Anti-de Sitter space (AdS$_3$). In relation to its parent theory, the mass of the extra degree of freedom controls the size of the fiber.

An appealing aspect of warped black holes is their delicate balance between simplicity and complexity. They are simple configurations because they are a quotient of a warped AdS$_3$ spacetime [8]: this places them on a similar footing to the BTZ black hole [9,10], and several concepts that are useful in BTZ can be applied to warped black holes [11]. Their complexity is due to its warped nature: a warped spacetime is neither locally, nor asymptotically, AdS$_3$ which makes it an instance of non-AdS holography. Moreover, this deviation from AdS has similarities with the near horizon geometry of the extreme Kerr black hole [12]. This places several holographic aspects of warped solutions closer to the challenges faced by Kerr/CFT [13], where it remains difficult to construct (or even characterise!) precisely the field theory that would represent a holographic dual in Kerr/CFT.

Our aim here is to differentiate among different proposals of a holographic dual to warped black holes. At the moment there are at least three different proposals to describe them holographically. Based on the results in [8], one expectation is that warped spacetimes (WAdS) are dual to a two-dimensional conformal field theory (CFT$_2$). This gives rise to a WAdS/CFT$_2$ duality, and some evidence towards it includes [14–17]. Another approach is to view the dual to warped spaces as a CFT$_2$ for which one turns on a suitable irrelevant operator. The choice of deformation is such that the theory becomes non-relativistic, and in particular one would break the conformal group from $sl(2) \times sl(2)$ down to $sl(2) \times u(1)$. Two approaches that can ac-

complish this mechanism are either a dipole deformation [18,19] or a J$\bar{\text{T}}$ deformation [20,21].

Here we will take a third approach, where the proposed dual theory to WAdS is expected to be a warped conformal field theory (WCFT). In its essence, a WCFT is a non-relativistic field theory whose symmetries are $sl(2) \times u(1)$. Examples of WCFTs, and its field theoretic properties, have been reported in [22–29], and evidence towards a WAdS/WCFT correspondence can be found in [11,30–36]. Although this proposal seems compelling, and might be compatible with the second proposal involving deformations, it also suffers from ambiguities. As observed in [11], a WCFT seems to admit two different facades: a description in terms of a *canonical* ensemble or a *quadratic* ensemble.[1] The main difference between these ensembles is a choice of coordinates. This coordinate transformation is state-dependent, and was introduced in [11] to make a WCFT mimic some thermodynamic properties of a CFT$_2$. In this context, the canonical ensemble is a natural choice to describe the non-relativistic system using a state-independent algebra; the quadratic ensemble has a state-dependent algebra that tries to imitate a CFT$_2$. We will review the definitions and properties of each of these ensemble in the next section.

The notion of ensembles also percolates into the definition of a warped black hole, giving rise to a canonical ensemble solution and its counterpart quadratic ensemble black hole. In this work we will be able to distinguish the fitness of each ensemble at setting up a holographic dictionary between WAdS and WCFT, and we will also comment on the WAdS/CFT$_2$ proposal. Our approach exploits the near-extremal limit of warped black holes, which will be used as a lamppost to establish the basic features of a holographic dual. Extremality corresponds to the zero temperature black hole, where the inner and outer horizon coincide. Near-extremality splits apart these horizons slightly: this increases the temperature by a small degree, and it induces an increase of mass and entropy (and although not essential, angular momentum is kept fixed in our analysis). Taking a near-extremal limit is a useful strategy. As it has been advocated in [37,38], and shown in countless examples,[2] the near-extremal dynamics of a black hole is well approximated by Jackiw-Teitelboim (JT) gravity [40,41]. This provides a universal sector in the low-temperature regime of the black hole, which can capture both classical and quantum aspects of the black hole as one ignites the solutions from extremality to near-extremality.

When we apply these new developments to warped black holes, we will see that each ensemble (canonical and quadratic) follows parallel and consistent descriptions at the classical level. In the near-extremal limit, we will analyse the thermodynamic properties of their Wald entropy, the properties of the near-horizon geometry and correlation functions. We will also construct a low-energy (IR) effective theory that describes the near-extremal dynamics: this theory contains a JT sector, in addition to a massive degree of freedom. All these quantities can be mapped and contrasted using the state-dependent coordinate transformation without any issues, and we find perfect agreement among the quantities considered here. The remarkable results come from the fact that the IR theory and the dual WCFT[3] *independently* make predictions about quantum corrections to the black hole entropy in the near-extremal regime. Comparing the quantum corrections to the entropy predicted by these derivations gives a non-trivial test to WAdS/WCFT: only the canonical ensemble is compatible with the prediction of the effective IR description. We deem this as a non-trivial and compelling reason to discard the quadratic ensemble as a description of quantum properties of warped black holes.

The analysis of the near-extremal dynamics of warped black holes will be done when they are solutions to topologically massive gravity. Regardless of the theory used, we expect that qualitatively the observables involved in our analysis will be robust and follow the trend de-

---

[1]The word "ensemble" is used here to match the nomenclature in [11]. It denotes a frame, or set of variables, to describe the theory; it has nothing to do with ensembles in statistical physics.

[2]For a recent review, see [39].

[3]The field-theoretic analysis of the near-extremal limit of WCFTs was done in [28,29].

scribed here; this is due to the robustness of the Wald entropy and the fact that the quantities in play rely mainly on the geometrical properties of the background and not the theory. In particular, from the perspective of the IR effective theory, the appearance of a JT sector will be universal. However, we expect differences to arise which involve the additional massive degree of freedom in the IR theory. In our analysis, it will appear as a massive scalar field, which has a negative mass squared. There is a range for which the field is stable, and would be dual to a relevant operator. However, it can also create an instability in the theory—the same one found in [42]. We suspect this instability is specific to topologically massive gravity. We will comment more on these differences in our final section.

This paper is structured as follows. We start in Sec. 2 with a review of warped black holes as solutions to three-dimensional topologically massive gravity. In this context, we introduce the canonical and quadratic ensembles, and for each ensemble we overview their thermodynamic properties at finite temperature, the associated asymptotic symmetry group, and how these quantities are compatible with a dual WCFT description. The next two sections, Sec. 3 and Sec. 4, we cover several aspects of the near-extremal limit of warped black holes. The content is presented in a way that it treats in parallel the properties of canonical and quadratic ensemble of warped black holes. For each ensemble we report on the extremal limit, the low temperature response of thermodynamic variables, the near-horizon geometry at near-extremality, and the behaviour of greybody factors in the near-extremal regime. In Sec. 5 we take a different approach to the near-extremal limit: via dimensional reduction, we construct an effective description of the near-AdS$_2$ region. This effective IR theory should consistently describe the response of the black hole due to turning on a small temperature at fixed angular momentum. As a simple check, we verify that the solutions in Sec. 3.2 and Sec. 4.2 are correctly captured by the effective IR theory. In Sec. 6 we discuss and contrast warped black holes from various perspectives. We first contrast the results in Sec. 3 and Sec. 4. Then we contrast those to the near-extremal limit of WCFT. And finally we contrast with the outcomes of the near-AdS$_2$ theory. We conclude with a summary of our main findings and outlook in Sec. 7.

## 2 Black holes in topologically massive gravity

In this section we review the basic features of the two families of black holes we will be considering in this work. These fall under the broad umbrella of warped black holes (WBH), with one family denoted as black holes in the *canonical ensemble* (CE) and the other as black holes in the *quadratic ensemble* (QE). They share several similarities and ties, which we will highlight below, and also stress their differences.

One particularly interesting gravitational theory in three dimensions in which these WBH can be embedded is topologically massive gravity (TMG) [43–45]. In terms of its action, TMG contains two terms: the Einstein-Hilbert action and a gravitational Chern-Simons term. The explicit expression is

$$I_{\text{3D}} = I_{\text{EH}} + I_{\text{CS}}, \tag{1}$$

where the two contributions are

$$
\begin{aligned}
I_{\text{EH}} &= \frac{1}{16\pi G_3} \int d^3x \sqrt{-g} \left( \mathscr{R}^{(3)} - 2\Lambda \right), \\
I_{\text{CS}} &= \frac{1}{32\pi G_3 \mu} \int d^3x \sqrt{-g} \, \varepsilon^{MNL} \left( \Gamma^P_{MS} \partial_N \Gamma^S_{LP} + \frac{2}{3} \Gamma^P_{MS} \Gamma^S_{NQ} \Gamma^Q_{LP} \right),
\end{aligned}
\tag{2}
$$

where $\mathscr{R}^{(3)}$ denotes the three-dimensional Ricci scalar. We have added a cosmological constant to the Einstein-Hilbert term, which we will take to always be negative: $\Lambda = -1/\ell^2$. The

gravitational Chern-Simons term is controlled by a real coupling $\mu$ that has dimensions of mass. The equations of motion of TMG are

$$\mathcal{R}^{(3)}_{MN} - \frac{1}{2} g_{MN} \mathcal{R}^{(3)} - \frac{1}{\ell^2} g_{MN} = -\frac{1}{\mu} C_{MN}\,, \tag{3}$$

where $C_{MN}$ is the Cotton tensor,[4]

$$C_{MN} = \epsilon_M{}^{QP} \nabla_Q \left( \mathcal{R}^{(3)}_{PN} - \frac{1}{4} g_{PN} \mathcal{R}^{(3)} \right)\,. \tag{4}$$

To categorise the solutions to TMG, it is common to introduce the dimensionless coupling

$$\nu \equiv \frac{\mu \ell}{3}\,. \tag{5}$$

Without loss of generality, we will always take $\nu$ to be a positive number. This can always be reverted by a choice of orientation, since the Chern-Simons action is parity odd.

There are two branches of solutions in TMG that will be recurrent in our analysis:

**Warped backgrounds.** These solutions have a non-vanishing Cotton tensor, $C_{MN} \neq 0$. The term warped is used to highlight the symmetries of the vacuum solutions of this theory: warped AdS$_3$ (WAdS$_3$). These are the most simple non-Einstein manifolds one can obtain in TMG, and we will review their properties below. In this context, we will focus on the so-called warped black hole solutions [1, 2, 8], which are quotients of specific instances of WAdS$_3$.

**Locally AdS$_3$ backgrounds.** These have a vanishing Cotton tensor, $C_{MN} = 0$, and hence the backgrounds are independent of $\mu$. These solutions are also part of the classical phase space of pure AdS$_3$ gravity, i.e., when only the Einstein-Hilbert term is in play. And as it is well-known, these type of solutions are all locally AdS$_3$ spacetimes. Among this class, the solution that will be prominently used here is the BTZ black hole [9, 10] as a means to contrast against the warped black holes.

It is worth reviewing in more detail the general properties of WAdS$_3$. We will be following the discussion in [8]. Similar to AdS$_3$, the warped solution is a real line fibration over AdS$_2$, with the crucial difference being that the size of the fibration of WAdS$_3$ depends on $\mu\ell$. This has the effect of breaking the $SO(2,2)$ symmetries of AdS$_3$ down to $SL(2,\mathbb{R}) \times U(1)$. In this context there are three categories of vacua: spacelike, timelike, and null WAdS$_3$. This nomenclature refers to the signature of the fibration. For spacelike and timelike vacua, there are two distinct cases depending on $\nu$: stretched ($\nu^2 > 1$), and squashed ($\nu^2 < 1$). Null WAdS$_3$ requires that $\nu = 1$, and there are two choices for the sign of the fiber.

For our work, the relevant vacua are *spacelike* and *timelike* WAdS$_3$. Spacelike WAdS$_3$ is given by the metric

$$ds^2 = \frac{\ell^2}{\nu^2 + 3} \left( -\cosh^2 \sigma \, d\tau^2 + d\sigma^2 + \frac{4\nu^2}{\nu^2 + 3} (du - \sinh \sigma \, d\tau)^2 \right)\,, \tag{6}$$

with $\{\tau, \sigma, u\} \in [-\infty, \infty]$. In this coordinate system the structure of the fiber and isometries of the vacua are manifest. Note that for $\nu = 1$ one recovers an AdS$_3$ space with $SO(2,2)$ isometries.[5] The warped black holes we will study here are obtained as quotients of this space, but this requires that $\nu^2 \geq 1$ in order to avoid closed timelike curves [8].

---

[4]We are using convention where $\sqrt{-g}\,\epsilon^{012} = -1$. Indices with capital latin letters label three-dimensional spacetime, i.e., $M, N, \ldots = \{0, 1, 2\}$.

[5]Note that while being smoothly connected to WAdS$_3$, the AdS$_3$ vacuum and the locally AdS$_3$ backgrounds discussed above exist as classical solutions regardless the value of $\nu$.

The timelike WAdS$_3$ metric is given by

$$ds^2 = -dt^2 + \frac{dr^2}{r\big((v^2+3)r+4\big)} - 2vr dt d\phi + \frac{r}{4}\big(3(1-v^2)r+4\big)d\phi^2, \qquad (7)$$

with $\phi \sim \phi + 2\pi$. These coordinates cover the global spacetime, and for $v > 1$, there are closed timelike curves at large $r$. Alternatively, global timelike WAdS$_3$ can be obtained from (6) by taking $u \to i\tau$, $\tau \to iu$. It will appear when we review the properties of the vacuum state in the holographic picture.

## 2.1 Warped black hole: Canonical ensemble

The first class of black hole solutions will be referred to as WAdS$_3$ black holes in the canonical ensemble. These appeared in [1, 2, 30, 46, 47] and were first studied holographically in [8]. The metric is given by

$$ds^2 = -N(r)^2 dt^2 + \frac{\ell^2}{4R(r)^2 N(r)^2}dr^2 + R(r)^2\big(d\theta - N^\theta(r)dt\big)^2, \qquad (8)$$

where we defined

$$\begin{aligned}
R(r)^2 &= \frac{r}{4}\Big(3(v^2-1)r + (v^2+3)(r_+ + r_-) - 4v\sqrt{r_+ r_-(v^2+3)}\Big), \\
N(r)^2 &= \frac{1}{4R(r)^2}(v^2+3)(r-r_+)(r-r_-), \\
N^\theta(r) &= \frac{2vr - \sqrt{r_+ r_-(v^2+3)}}{2R(r)^2}.
\end{aligned} \qquad (9)$$

The constants $r_\pm$ determine the positions of the outer and inner horizons of the black hole. These solutions are obtained as a discrete quotient from the metric in (6), much like BTZ black holes are discrete quotients of global AdS$_3$. Actually for $v = 1$, the metrics are locally AdS$_3$, and represent BTZ black holes, albeit in an unusual coordinate system. A difference though is that the global metric (6) is not recovered for any value of the black hole parameters [42]. However, for the choice

$$r_+ = -\frac{4i\ell}{v^2+3}, \quad r_- = 0, \qquad (10)$$

the metric possesses enhanced symmetries (four Killing vectors forming the algebra of $sl(2,\mathbb{R}) \times u(1)$, instead of two generically). The metric is then complex, and the analytic continuation $r \to ir$, $t \to -it$ brings it to the global timelike WAdS$_3$ metric (7), which is viewed as the global vacuum [11].[6]

These black holes satisfy the usual thermodynamics laws, which we now review. The mass $M^{\text{CE}}$ and angular momentum $J^{\text{CE}}$ of the black hole, are defined as conserved charges associated to $\partial_t$ and $\partial_\theta$ respectively, and are given by

$$\begin{aligned}
M^{\text{CE}} &= \frac{v^2+3}{24vG_3\ell}\Big((r_+ + r_-)v - \sqrt{(v^2+3)r_+ r_-}\Big), \\
J^{\text{CE}} &= \frac{(v^2+3)}{96vG_3\ell}\left[\Big((r_+ + r_-)v - \sqrt{(v^2+3)r_+ r_-}\Big)^2 - \frac{5v^2+3}{4}(r_+ - r_-)^2\right].
\end{aligned} \qquad (11)$$

---

[6]Notice that the determination of the vacuum metric is ambiguous. First, $r_+$ and $r_-$ could be switched. Second, the choice (10) is not unique as in the AdS$_3$ situation. For instance, symmetry enhancement from two to four Killing vectors occurs for $r_+ - r_- = -\frac{4i\ell}{3+v^2}$.

These values are tied to the theory the WBH belongs to: TMG in this case. They are also tied to the black hole background: this is why we are using the superscript "CE", which stands for canonical ensemble. The Wald entropy of the black hole (8) is given by [8, 48–50]

$$S^{\text{CE}} = \frac{\pi}{24\nu G_3}\left((9\nu^2+3)r_+ - (\nu^2+3)r_- - 4\nu\sqrt{(\nu^2+3)r_+r_-}\right). \tag{12}$$

The first law of black hole thermodynamics then reads

$$dM^{\text{CE}} = T^{\text{CE}}dS^{\text{CE}} + \Omega^{\text{CE}}dJ^{\text{CE}}, \tag{13}$$

where the Hawking temperature and angular velocity are given by

$$T^{\text{CE}} = \frac{\nu^2+3}{4\pi\ell}\frac{r_+ - r_-}{2\nu r_+ - \sqrt{(\nu^2+3)r_+r_-}},$$
$$\Omega^{\text{CE}} = \frac{2}{\left(2\nu r_+ - \sqrt{(\nu^2+3)r_+r_-}\right)}. \tag{14}$$

The thermodynamic behaviour of a WBH can be accounted for holographically by making use of the symmetries of their semi-classical phase space. The key observations are as follows. A phase space accommodating the WBH solutions (but *not* the global timelike WAdS$_3$ vacuum) was proposed and further studied in [30–33]. Its symmetry algebra is generated by the following asymptotic Killing vectors,

$$\ell_n = e^{in\theta}\partial_\theta - inre^{in\theta}\partial_r, \qquad p_n = e^{in\theta}\partial_t, \tag{15}$$

with $n \in \mathbb{Z}$. To each of these vectors we can associate a corresponding conserved charge, which we denote as $L_n$ and $P_n$. In particular, the zero modes are related to the mass and angular momentum in (11) via

$$P_0 = M^{\text{CE}}, \qquad L_0 = -J^{\text{CE}}. \tag{16}$$

The algebra for the charges is given by

$$[L_n, L_m] = (n-m)L_{n+m} + \frac{c}{12}(n^3-n)\delta_{n+m},$$
$$[P_n, P_m] = \frac{\mathsf{k}}{2}n\delta_{n+m},$$
$$[L_n, P_m] = -mP_{m+n}. \tag{17}$$

This is a Virasoro-Kac-Moody algebra. The central extensions appearing here that are appropriate for TMG are given by [31]

$$c = \frac{(5\nu^2+3)}{\nu(\nu^2+3)}\frac{\ell}{G_3}, \qquad \mathsf{k} = -\frac{\nu^2+3}{6\nu}\frac{1}{\ell G_3}. \tag{18}$$

In the same way that the Virasoro algebra incarnates the symmetries of a CFT$_2$, the algebra (17) represents those of a warped CFT [11], that is a two-dimensional field theory with chiral scaling

$$\theta \to f(\theta), \qquad t \to t + g(\theta). \tag{19}$$

Here $f(\theta)$ is a diffeomorphism and $g(\theta)$ an arbitrary function. This suggests that a gravity theory with WAdS$_3$ boundary conditions is dual to a WCFT, whose intrinsic and holographic properties have been explored in various works [11, 22–36, 51–65].

WCFTs are able to capture certain properties of WAdS$_3$ backgrounds [7, 11] and of higher dimensional spacetimes [4, 66, 67]. Here we will highlight how the black hole mechanics can be reproduced by the thermodynamic behaviour of a WCFT. Using an adapted form of the Cardy formula, one can show that at high-temperature the entropy of a WCFT is given by [11]

$$S^{\text{WCFT-CE}} = -\frac{4\pi i P_0 P_0^{\text{vac}}}{k} + 4\pi \sqrt{-\left(L_0^{\text{vac}} - \frac{(P_0^{\text{vac}})^2}{k}\right)\left(L_0 - \frac{P_0^2}{k}\right)} \, . \tag{20}$$

In order to compare this expression with $S^{\text{CE}}$ in (12), we need to provide the values of $L_0^{\text{vac}}$ and $P_0^{\text{vac}}$, i.e., determine the vacuum charges. One key obstacle, relative to a CFT$_2$, is that for a WCFT the vacuum charges are not fully specified by the symmetries alone. Under the assumption that the vacuum state is normalizable and invariant under $sl(2,\mathbb{R}) \times u(1)$, one can show that

$$L_0^{\text{vac}} = -\frac{c}{24} + \frac{(P_0^{\text{vac}})^2}{k} \, . \tag{21}$$

It is interesting to note that (10) satisfies this relation. Therefore, taking (10) as a choice of vacuum state, and using (11), we would have

$$L_0^{\text{vac}} = -\frac{1}{24\nu}\frac{\ell}{G_3} \, , \qquad P_0^{\text{vac}} = -\frac{i}{6}\frac{1}{G_3} \, . \tag{22}$$

It is straightforward to check that $S^{\text{CE}} = S^{\text{WCFT-CE}}$, provided (18), (16) and (22).

The values in (18) and (22) show a persistent feature of holographic WCFTs: they typically have negative level and $P_0^{\text{vac}}$ is purely imaginary. This makes the theories non-unitary; still, these features are manageable, rich, and interesting as they lead to intriguing synergy with black holes [27, 29].

## 2.2 Warped black hole: Quadratic ensemble

Another family of spacetimes we will be considering are the so-called Warped BTZ metrics; also known as WBHs in the quadratic ensemble, a nomenclature that will become clear below. Their line element reads

$$ds^2 = -N_{\text{QE}}(r)^2 dt^2 + \frac{(1-2H^2)}{R_{\text{QE}}(r)^2 N_{\text{QE}}(r)^2}r^2 dr^2 + R_{\text{QE}}(r)^2\left(d\varphi + N^\varphi(r)dt\right)^2 \, , \tag{23}$$

where

$$R_{\text{QE}}(r)^2 = (1-2H^2)r^2 - 2H^2\frac{(r^2-r_+^2)(r^2-r_-^2)}{(r_+ + r_-)^2} \, ,$$

$$N_{\text{QE}}(r)^2 = \frac{(1-2H^2)}{R_{\text{QE}}(r)^2 L^2}(r^2-r_+^2)(r^2-r_-^2) \, , \tag{24}$$

$$N^\varphi(r) = -\frac{1}{R_{\text{QE}}(r)^2 L}\left((1-2H^2)r_+ r_- + 2H^2\frac{(r^2-r_+^2)(r^2-r_-^2)}{(r_+ + r_-)^2}\right) \, .$$

Here $r_\pm$ are constants, which determine the positions of the outer and inner horizon. $H^2$ and $L$ are related to the TMG parameters $\nu = \mu\ell/3$ and $\ell$ through

$$H^2 = -\frac{3(\nu^2-1)}{2(\nu^2+3)} \, , \qquad L = \frac{2\ell}{\sqrt{\nu^2+3}} \, . \tag{25}$$

Notice that $1-2H^2 = \nu^2 L^2/\ell^2$. The above metric can be viewed as a deformation of the BTZ black hole with AdS$_3$ radius $L$. That is, the line element (23) is equivalent to

$$ds^2 = ds^2_{\text{BTZ}} - 2H^2\zeta \otimes \zeta \, , \tag{26}$$

where $\zeta = \frac{1}{r_+ + r_-}(-L\partial_t + \partial_\varphi)$ is a spacelike Killing vector of unit norm and $H^2$ determines the deviation away from the BTZ black hole. For $H^2 = 0$, we recover the BTZ black holes in usual coordinates, with AdS$_3$ radius $L$.

The mass and angular momentum of this black hole are given by

$$M^{\text{QE}} = \frac{(3-4H^2)(r_+^2 + r_-^2) - 2r_- r_+}{24 G_3 L \sqrt{1-2H^2}},$$
$$J^{\text{QE}} = \frac{r_+^2 + r_-^2 - 2(3-4H^2) r_+ r_-}{24 G_3 L \sqrt{1-2H^2}}, \tag{27}$$

and the Wald entropy is

$$S^{\text{QE}} = \frac{\pi}{6 G_3 \sqrt{1-2H^2}}((3-4H^2) r_+ - r_-). \tag{28}$$

Here the superscript "QE" denotes quadratic ensemble. As expected these quantities satisfy a first law, which reads

$$dM^{\text{QE}} = T^{\text{QE}} dS^{\text{QE}} + \Omega^{\text{QE}} dJ^{\text{QE}}, \tag{29}$$

where the Hawking temperature and angular velocity are

$$T^{\text{QE}} = \frac{r_+^2 - r_-^2}{2\pi L r_+}, \qquad \Omega^{\text{QE}} = -\frac{r_-}{r_+}. \tag{30}$$

As we did in Sec. 2.1, we will next review how to capture the thermodynamic properties of the quadratic ensemble solution holographically. Starting with the phase space, boundary conditions containing the WBH in the quadratic ensemble, and their symmetry algebra, have been identified in [36]. The asymptotic vector fields that enter in this construction, to leading order, are given by

$$\tilde{\ell}_n = e^{inx^+}(\partial_+ - \frac{1}{2} inr\partial_r), \qquad \tilde{p}_n = e^{inx^+}\partial_-, \tag{31}$$

with $x^\pm = \frac{t}{L} \pm \varphi$ and $n \in \mathbb{Z}$. Each of these generators has an associated conserved charge, which we coin $\mathscr{L}_n$ and $\mathscr{P}_n$. The relation of the zero modes ($n = 0$) to the mass and angular momentum is

$$M^{\text{QE}} = \frac{1}{L}(\mathscr{L}_0 + \mathscr{P}_0),$$
$$J^{\text{QE}} = \mathscr{L}_0 - \mathscr{P}_0. \tag{32}$$

The corresponding charge algebra is then found to be

$$[\mathscr{L}_n, \mathscr{L}_m] = (n-m)\mathscr{L}_{n+m} + \frac{c}{12}(n^3 - n)\delta_{n+m},$$
$$[\mathscr{L}_n, \mathscr{P}_m] = -m\mathscr{P}_{m+n}, \tag{33}$$
$$[\mathscr{P}_n, \mathscr{P}_m] = -2n\mathscr{P}_0 \delta_{m+n},$$

with central charge $c$ given by (18), which in terms of the variables used here is

$$c = \frac{2(1-H^2)}{\sqrt{1-2H^2}} \frac{L}{G_3}. \tag{34}$$

The algebra (33) resembles a Virasoro-Kac-Moody algebra, but with a key twist. The level of the affine $u(1)$ generators is controlled by $\mathscr{P}_0$: this makes the algebra non-local. Another curious, and useful, observation is that the algebras (17) and (33) are related though the redefinition

$$\mathscr{L}_n = L_n - \frac{2}{\text{k}} P_0 P_n + \frac{1}{\text{k}} P_0^2 \delta_n, \quad \mathscr{P}_n = -\frac{2}{\text{k}} P_0 P_n + \frac{1}{\text{k}} P_0^2 \delta_n. \tag{35}$$

This transformation is the reason why we refer to solutions in this classical phase space as being in a "quadratic ensemble."

Despite the undesirable non-local aspects it is possible to extract information about the density of states for Hilbert spaces described by (33). As shown in [11], starting from the partition function

$$Z^{\text{WCFT-QE}}(\beta_R, \beta_L) = \text{Tr } e^{-\beta_R \mathscr{P}_0 - \beta_L \mathscr{L}_0}, \tag{36}$$

it is possible to extract a universal behaviour at high temperatures, analogous to the Cardy behaviour in a $\text{CFT}_2$. More concretely, the relation (35) allows to relate properties of (36) to those of a regular WCFT, which leads to the following entropy formula

$$S^{\text{WCFT-QE}} = 4\pi \sqrt{-\mathscr{P}_0^{\text{vac}} \mathscr{P}_0} + 4\pi \sqrt{-\mathscr{L}_0^{\text{vac}} \mathscr{L}_0}. \tag{37}$$

Despite the absence of full conformal invariance of the system, one obviously cannot help but notice the similarity between (37) and the Cardy formula of a $\text{CFT}_2$; but we stress that $\mathscr{P}_0^{\text{vac}}$ and $\mathscr{L}_0^{\text{vac}}$ are not fixed by symmetries. One simple, and interesting, check is to notice that $S^{\text{WCFT-QE}}$ is equivalent to $S^{\text{WCFT-CE}}$ in (20), due to (35).

We can now proceed to compare (37) to the Wald entropy (28). The key is to choose a vaccum state. We will use (10) and (35); with this we infer that in the quadratic ensemble the vacuum charges are

$$\mathscr{L}_0^{\text{vac}} = -\frac{c}{24}, \qquad \mathscr{P}_0^{\text{vac}} = \frac{1}{36 \mathsf{k} G_3^2}, \tag{38}$$

where $c$ and $\mathsf{k}$ are defined in (18). With this choice, and using (32), it is simple to check that $S^{\text{WCFT-QE}} = S^{\text{QE}}$. In this comparison, it is also useful to report how the potentials (30) are related to WCFT variables. We have

$$\frac{1}{T^{\text{QE}}} = \frac{1}{2}(\beta_R + \beta_L), \qquad \frac{\Omega^{\text{QE}}}{T^{\text{QE}}} = \frac{1}{2}(\beta_L - \beta_R), \tag{39}$$

where the left and right moving potentials have a very simple expression,

$$\beta_L = \frac{2\pi L}{r_+ - r_-}, \qquad \beta_R = \frac{2\pi L}{r_+ + r_-}. \tag{40}$$

## 2.3 Ties between canonical and quadratic ensemble

Until now we have been treating (8) and (23) as two distinct black hole solutions of TMG. Here we will review how they are intimately related, and fit it with the relation among the generators in (35). The basic observation is that the metrics are related by the following change of coordinates[7]

$$\begin{aligned}
\frac{t}{L} &= -\frac{\mathsf{k}}{4M^{\text{CE}}} t, \\
\varphi &= \theta + \frac{\mathsf{k}}{4M^{\text{CE}}} t, \\
r^2 &= \frac{(\nu^2 + 3)}{4\nu^2}\left(R(r)^2 - \frac{3}{4}(\nu^2 - 1)(r - r_+)(r - r_-)\right),
\end{aligned} \tag{41}$$

where

$$M^{\text{CE}} = -\frac{\mathsf{k}}{4}\left((r_+ + r_-)\nu - \sqrt{(\nu^2 + 3)r_+ r_-}\right), \tag{42}$$

---

[7]The radial component of the diffeomorphism is such that the radial function in (9) and (24) report the same value, i.e., $R(r)^2 = R_{\text{QE}}(r)^2$. For more details see [68].

which is just a re-writing of (11) in terms of the level k. Since the mass, $M^{\mathrm{CE}}$, enters here this is a state-dependent transformation between the two solutions.

The coordinate transformation would also lead to the relation between the generators (35). We also note that $S^{\mathrm{CE}} = S^{\mathrm{QE}}$, that is the entropy of the CE and QE WBH match; this is expected from the perspective of the WCFT, and also it is expected since the Wald entropy is diffeomorphism invariant. It will be useful to record for later derivations the relations between the thermodynamic potentials, which reads

$$\beta^{\mathrm{CE}}\Omega^{\mathrm{CE}} = \beta_L, \qquad \beta^{\mathrm{CE}} = -\frac{2\pi}{3\mathsf{k}G_3}\left(1 + \frac{\beta_R}{\beta_L}\right). \tag{43}$$

Here $\beta^{\mathrm{CE}} = 1/T^{\mathrm{CE}}$ and the potentials are defined in (14) and (40) for each ensemble. At first glance the diffeomorphism (41) seems trivial and should indicate that the dual description in the canonical and quadratic ensemble is simply a choice. However we also have reviewed that the implications of this transformation gives a non-local algebra in one case, which is dramatic. In the following sections our task will be to analyse and contrast the solutions starting from their near-extremal limit. With this we aim to decode differences and similarities among these two ensembles.

## 3 Near-extremal warped black holes: Canonical ensemble

Our analysis starts with the WBH solution, casted in the canonical ensemble (CE). Building on the general features reviewed in the previous section, we will focus on three aspects of the solution near-extremality: the response of thermodynamic quantities, the shape of the near horizon geometry, and the scattering of massive scalar fields. For BTZ black holes, these aspects have been addressed in various works including [69–72].

### 3.1 Thermodynamics

An important aspect of our work is to consider the extremal version of the metrics (8), and look at small deviations away from it. In the following we will introduce these concepts for the CE warped black hole and define the concept of "near-extremality" from the thermodynamic perspective.

The extremal black hole is defined as the solution of (8) for which

$$\textbf{Extremality:} \quad r_+ = r_- \equiv r_0. \tag{44}$$

It is important to remark that the potentials (14) are only well defined in this limit if in addition $\nu \neq 1$. This means that the extremal CE solution is not smoothly connected to the extremal BTZ black hole. For this reason, we stress that in the following equations we always assume $\nu > 1$. At extremality, the potentials (14) take the values

$$\begin{aligned}
T^{\mathrm{CE}}\Big|_{r_\pm = r_0} &= 0, \\
\Omega^{\mathrm{CE}}\Big|_{r_\pm = r_0} &= \frac{2}{\left(2\nu - \sqrt{\nu^2 + 3}\right)r_0} \equiv \Omega^{\mathrm{CE}}_{\mathrm{ext}},
\end{aligned} \tag{45}$$

while the charges (11) become

$$
\begin{aligned}
M_{\text{ext}}^{\text{CE}} &\equiv M^{\text{CE}}\Big|_{r_\pm = r_0} = \frac{v^2 + 3}{12\, v\ell\, G_3}\frac{1}{\Omega_{\text{ext}}^{\text{CE}}}\,, \\
J_{\text{ext}}^{\text{CE}} &\equiv J^{\text{CE}}\Big|_{r_\pm = r_0} = \frac{v^2 + 3}{24\, v\ell\, G_3}\frac{1}{(\Omega_{\text{ext}}^{\text{CE}})^2}\,, \\
S_{\text{ext}}^{\text{CE}} &\equiv S^{\text{CE}}\Big|_{r_\pm = r_0} = \frac{\pi}{3 G_3}\frac{1}{\Omega_{\text{ext}}^{\text{CE}}}\,.
\end{aligned}
\tag{46}
$$

Near extremality is a small deviation from extremality leading to a non-vanishing temperature, while keeping the angular momentum $J_{\text{ext}}^{\text{CE}}$ fixed.[8] This can be achieved by modifying (44) as

$$
\textbf{Near-extremality:}\quad r_+ = r_0 + \epsilon\,\mathfrak{d}\,,\quad r_- = r_0 - \epsilon\,\mathfrak{d}\,.
\tag{47}
$$

Here $\epsilon \ll 1$, that is a small parameter that introduces the small deviation from extremality; $\mathfrak{d}$ is a parameter that will remain fixed as one takes $\epsilon \to 0$ and was chosen to keep $J_{\text{ext}}^{\text{CE}}$ fixed at leading order in $\epsilon$. The leading order response in $\epsilon$ of the temperature is linear, and reads

$$
T^{\text{CE}} = \frac{v^2 + 3}{4\pi\ell}\Omega_{\text{ext}}^{\text{CE}}\,\mathfrak{d}\,\epsilon + \mathcal{O}(\epsilon^2)\,.
\tag{48}
$$

The mass on the other hand increases quadratically

$$
\Delta E^{\text{CE}} = M^{\text{CE}} - M_{\text{ext}}^{\text{CE}} = \frac{(T^{\text{CE}})^2}{M_{\text{gap}}^{\text{CE}}} + \mathcal{O}(\epsilon^3)\,,
\tag{49}
$$

where the commonly coined mass gap [73–75] is given by

$$
M_{\text{gap}}^{\text{CE}} \equiv \frac{6 G_3}{\pi^2 \ell}\frac{v(3 + v^2)}{(3 + 5 v^2)}\Omega_{\text{ext}}^{\text{CE}} = \frac{3}{\pi^2 c}\sqrt{\frac{-\mathsf{k}}{J_{\text{ext}}^{\text{CE}}}}\,.
\tag{50}
$$

In the last equality we used (18) and (46) to cast the mass gap in terms of the central extensions that enter in the holographic description. Note that since $v > 1$ ($\Omega_{\text{ext}}^{\text{CE}} > 0$) the mass gap is always positive.[9]

It then also follows that the entropy responds linearly in temperature as we deviate from extremality with the slope inversely proportional to the mass gap:

$$
S^{\text{CE}} = S_{\text{ext}}^{\text{CE}} + 2\frac{T^{\text{CE}}}{M_{\text{gap}}^{\text{CE}}} + \mathcal{O}(\epsilon^2)\,.
\tag{51}
$$

This is the universal response of the entropy based on general grounds: that is, the compliance of the Wald entropy to a first law of thermodynamics and that extremality only involves two-coincident horizons. In our subsequent sections we will compare this analysis to the QE warped black hole, provide a derivation of the entropy via a holographic analysis, and match it to near-extremal limits of WCFTs.

---

[8]In many setups, such as Reissner-Nordstrom or Myers-Perry black holes, the prescription is to keep the conserved charge that controls the AdS$_2$ radius fixed, for reasons that will be more transparent in Sec. 5.2. In three dimensions this is not necessary, but it facilitates the analysis to keep one of the conserved charges fixed.

[9]Also recall that $\mathsf{k} < 0$ (18), so the mass gap is real.

## 3.2 Decoupling limit

In this portion we will report on the geometrical effects of the near-extremal limit. In this context a WBH behaves similarly to their $AdS_3$ counterparts: at extremality the near horizon geometry develops an $AdS_2$ throat. Here we show this limiting procedure explicitly, and also incorporate the near-extremal contributions. This will quantify the notion of near-$AdS_2$ for the WBH in the canonical ensemble.

As it is common practice, we start by introducing a coordinate transformation catered to zoom into the horizon of the black hole. First, in the context of the near-extremal limit (47), we introduce a new coordinate system $(\rho, \tau, \psi)$ which redefines the coordinates $(r, t, \theta)$ used in (8). The transformation reads

$$
\begin{aligned}
r &= r_0 + \epsilon \left( e^{\rho/\ell_2} + \frac{\mathfrak{d}^2}{4} e^{-\rho/\ell_2} \right), \\
t &= 2R_0 \frac{\ell_2}{\ell} \frac{\tau}{\epsilon}, \\
\theta &= \psi + 2\frac{\ell_2}{\ell} \frac{\tau}{\epsilon},
\end{aligned}
\tag{52}
$$

where $\epsilon$ and $\mathfrak{d}$ are defined around (47). In this context, $\epsilon$ implements extremality, and it is also the decoupling parameter that takes us to the near-horizon region, i.e., near to $r \to r_0$; a finite value of $\mathfrak{d}$ quantifies a deviation away from extremality. We have also introduced two constants in (52) which are defined as

$$
\ell_2^2 \equiv \frac{\ell^2}{\nu^2 + 3}, \quad R_0 \equiv R(r_0) \Big|_{r_\pm = r_0} = \frac{r_0}{2} \left( 2\nu - \sqrt{\nu^2 + 3} \right). \tag{53}
$$

As we will see momentarily, $\ell_2$ is the $AdS_2$ radius. It is also worth remarking that $\Omega_{\text{ext}}^{\text{CE}} = R_0^{-1}$, which is just a coincidence for the CE warped black hole. More significantly, $R_0$ is the size of the extremal horizon, and controls the extremal Wald entropy in (46).

The near-horizon region is defined by using (52) on (8) and taking the limit $\epsilon \to 0$, while keeping all other parameters fixed. The resulting line element is

$$
\begin{aligned}
ds_{\text{CE}}^2 = \, & d\rho^2 - e^{2\rho/\ell_2} \left( 1 - \frac{\mathfrak{d}^2}{4} e^{-2\rho/\ell_2} \right)^2 d\tau^2 \\
& + R_0^2 \left( d\psi + \frac{2\nu}{R_0} \frac{\ell_2}{\ell} e^{\rho/\ell_2} \left( 1 + \frac{\mathfrak{d}^2}{4} e^{-2\rho/\ell_2} \right) d\tau \right)^2 + \mathcal{O}(\epsilon).
\end{aligned}
\tag{54}
$$

As expected the result is finite, resulting into a non-degenerate metric, referred to as *self-dual warped AdS$_3$ space* [8,76]. This is the warped version of the near-horizon geometry of extremal BTZ black holes – self-dual $AdS_3$ space [77] –, and a constant polar angle section of the NHEK geometry [12, 13]. The first line reflects that the near-horizon geometry contains an $AdS_2$ factor: for $\mathfrak{d} = 0$, it is $AdS_2$ in Poincare coordinates, while for $\mathfrak{d} \neq 0$ the metric is locally $AdS_2$.[10] The later is usually coined "near-$AdS_2$". The second line reflects that the total space time is a fibration of a circle over $AdS_2$. The resulting local symmetries of the near-horizon region is therefore $sl(2, \mathbb{R}) \times u(1)$.

As we further explore the holographic properties of this black hole, it will also be important to quantify how the solution responds to first order away from extremality. With some foresight to the subsequent sections, we will parametrize the first order response in $\epsilon$ as

$$
ds_{\text{CE}}^2 = \left( \bar{g}_{\mu\nu} + \epsilon\, h_{\mu\nu} \right) dx^\mu dx^\nu + \left( R_0^2 + \epsilon \mathcal{Y} \right) \left( d\psi + (\bar{A}_\mu + \epsilon \mathcal{A}_\mu) dx^\mu \right)^2 + \cdots, \tag{55}
$$

---

[10]More specifically it is a Rindler (thermal) patch of $AdS_2$, where $\mathfrak{d}$ controls the acceleration of the observer. This geometry is also at times refered to as an "$AdS_2$ black hole."

that is, there is a response from the AdS$_2$ metric ($h_{\mu\nu}$), the size of the $U(1)$ circle ($\mathscr{Y}$), and the fiber ($\mathscr{A}_\mu$). Here the variables with a bar are those in (54): $\bar{g}_{\mu\nu}$ is the locally AdS$_2$ background, and $\bar{A}_\mu$ is the background component of the fibration. It is straightforward to read the values of these responses by keeping the first correction in $\epsilon$ of the coordinate transformation, where one finds

$$
\begin{aligned}
\mathscr{Y} &= 2\nu R_0\, e^{\rho/\ell_2}\left(1 + \frac{\eth^2}{4}e^{-2\rho/\ell_2}\right), \\
\mathscr{A} &= -\frac{\ell_2}{2\ell R_0^2}(3+5\nu^2)\,e^{2\rho/\ell_2}\left(1+\frac{\eth^4}{16}e^{-4\rho/\ell_2}\right)\mathrm{d}\tau - \frac{\eth^2}{R_0 r_0}\frac{\nu\ell_2}{\ell}\mathrm{d}\tau, \\
h_{\tau\tau} &= \frac{2\nu}{R_0}e^{3\rho/\ell_2}\left(1-\frac{\eth^2}{4}e^{-2\rho/\ell_2}\right)^2\left(1+\frac{\eth^2}{4}e^{-2\rho/\ell_2}\right), \quad h_{\rho\rho} = h_{\tau\rho} = 0.
\end{aligned}
\tag{56}
$$

As is persistent in many other black hole backgrounds, the responses grow rapidly as one reaches the boundary of AdS$_2$ at $\rho \to \infty$. This reflects that deviations away from extremality should be interpreted holographically as irrelevant deformations, and we will comment more on this in Sec. 5.

## 3.3 Two-point function

The behaviour of probes, and in particular its two-point function, is a useful way to encode properties of a black hole. Here we will analyse a massive probe around the CE warped black hole focusing on its near-extremal limit. The aim is to contrast the results against the analogous treatment for the BTZ black hole and the QE warped black hole; this comparison will be discussed in Sec. 6.1. Our derivations follow the analysis in, e.g., [72, 78].

We start by solving the Klein-Gordon equation of a scalar field with mass $m$ in the CE black hole background,

$$
\nabla^2\Phi(t, r, \theta) = m^2\Phi(t, r, \theta),
\tag{57}
$$

where $\nabla^2$ is the Laplace-Beltrami operator for the metric (8). Using a separable ansatz for $\Phi$, and further decomposing it into Fourier modes, we will write

$$
\Phi(t, r, \theta) = \sum_k \int \mathrm{d}\omega\, e^{-i\frac{\omega}{\ell}t + ik\theta}\Psi(r),
\tag{58}
$$

for which the wave equation then reads

$$
\begin{aligned}
\frac{\partial}{\partial r}\Big((r-r_+)(r-r_-)\frac{\partial}{\partial r}\Big)\Psi(r) \\
+ \frac{1}{\nu^2+3}\left(\frac{1}{N(r)^2}\big(\omega - N^\theta(r)\ell k\big)^2 - \frac{\ell^2 k^2}{R(r)^2}\right)\Psi(r) = \frac{\ell^2 m^2}{\nu^2+3}\Psi(r).
\end{aligned}
\tag{59}
$$

The functions $N(r)^2$, $R(r)^2$, and $N^\theta(r)$ are defined in (8). Note that we have chosen to normalise time in (58) with respect to $\ell$, which makes $\omega$ dimensionless.

Our main task in the following is to extract the two-point function by solving (59).[11] We are mainly interested in the behaviour near extremality, which implies that we are exploring the low-temperature and low-frequency limit of the correlation function. In this context the quantity that is interesting to report on is the relation between the two-point function evaluated in the UV region ($r \to \infty$) and the one evaluated in the IR region ($r \to r_+$). That is, the relation between the two-point function evaluated near the WAdS$_3$ boundary and the one in the near-AdS$_2$ region in Sec. 3.2.

---

[11]In our setups, a two-point function is equivalent to a greybody factor (up to an overall normalization).

To implement the near-extremal limit we will introduce very similar variables as in Sec. 3.2, with some small adjustments to avoid clutter. We define the dimensionless parameters

$$x \equiv \frac{r - r_+}{r_+}, \quad \tau_H \equiv \frac{r_+ - r_-}{r_+}. \tag{60}$$

Notice that near-extremality, as defined in (47), implies that $\tau_H \sim \epsilon \ll 1$. In terms of these variables, (59) becomes

$$\partial_x(x(x + \tau_H)\partial_x)\Psi(x) - \frac{4R_{\tau_H}^2}{\tau_H r_+^2} \frac{\ell_2^4}{\ell^4} \left( \omega - \frac{k\ell}{R_{\tau_H}} - \frac{\nu r_+}{R_{\tau_H}}\tau_H \omega \right)^2 \frac{1}{(x + \tau_H)}\Psi(x)$$
$$+ \frac{4R_{\tau_H}^2}{\tau_H r_+^2} \frac{\ell_2^4}{\ell^4} \left( \omega - \frac{k\ell}{R_{\tau_H}} \right)^2 \frac{1}{x}\Psi(x) - \ell_2^2 m_{\text{CE}}^2 \Psi(x) = 0, \quad (61)$$

where

$$R_{\tau_H} \equiv \frac{r_+}{2} \left( 2\nu - \sqrt{(\nu^2 + 3)(1 - \tau_H)} \right), \tag{62}$$

which is the non-extremal version of (53), and

$$m_{\text{CE}}^2 = m^2 + \frac{3\ell_2^2}{\ell^4}(1 - \nu^2)\omega^2, \tag{63}$$

with $\ell_2$ given by (53). $m_{\text{CE}}$ is an effective mass with a non-trivial frequency dependence—this is reminiscent of Kerr/CFT [78]. Note that in the BTZ limit, where $\nu = 1$, the frequency dependence drops from (63). It is also interesting to note that $m_{\text{CE}}$ enters in (61) measured in units of the AdS$_2$ radius, and not AdS$_3$.

To extract the two-point function, it is common to divide the wave equation into two zones,

$$\begin{aligned} \textbf{Far region:} \quad & x \gg \tau_H, \\ \textbf{Near region:} \quad & x \ll 1, \end{aligned} \tag{64}$$

as one takes $\tau_H \to 0$. The **far zone** reaches to the asymptotically warped AdS$_3$ portion of the geometry, far from the horizon of the black hole. The **near zone** covers the area close to the horizon, and near extremality, it corresponds to the near-AdS$_2$ portion of the geometry described in Sec. 3.2. In the near-extremal limit, these two regions overlap at

$$\textbf{Matching region:} \quad 1 \gg x \gg \tau_H. \tag{65}$$

As it is common in this sort of analysis, one solves the wave equation separately in the far and near region, and then overlaps them in the matching region. This gives a connection between the correlation functions in the UV (far) and IR (near) regimes.

For AdS$_3$ and WAdS$_3$, this matching procedure is very simple to implement. The reason being that the singularity structure in $x$ of the Klein-Gordon operator on (W)AdS$_3$ and AdS$_2$ is exactly the same. The difference between the near region and the whole geometry is the behaviour of the coefficients governing the poles. That is, the radial wave equation in the far, near and matching region has the general structure

$$\partial_x(x(x + \tau_H)\partial_x)\Psi(x) - \frac{\mathsf{a}(\omega, k)}{(x + \tau_H)}\Psi(x) + \frac{\mathsf{b}(\omega, k)}{x}\Psi(x) - \ell_2^2 m_{\text{CE}}^2 \Psi(x) = 0, \tag{66}$$

which is manifest in (61). The difference between each region arises from frequency dependence of $\mathsf{a}(\omega, k)$ and $\mathsf{b}(\omega, k)$; this reflects the details of an AdS$_2$ background versus (W)AdS$_3$,

which we will discuss in detail below.[12] The differential equation (66) can be solved exactly, and its solutions are governed by hypergeometric functions.

Within the generality of (66), we can report on the behaviour of the two-point function. In the far zone, where the variable $x$ is large, the solutions to the wave equation (66) reduces to

$$\Psi(x) = \psi_1(\omega, k)\, x^{\Delta_{\text{CE}}-1} + \psi_2(\omega, k)\, x^{-\Delta_{\text{CE}}}, \tag{67}$$

where

$$\Delta_{\text{CE}} \equiv \frac{1}{2} + \frac{1}{2}\sqrt{1 + 4\ell_2^2 m_{\text{CE}}^2}, \tag{68}$$

and $\psi_{1,2}$ are independent of $x$. Note that $\Delta_{\text{CE}}$ plays the role of a conformal dimension in AdS, although here it is frequency-dependent due to (63). We will impose in-going boundary conditions at the horizon, i.e., for $x \ll 1$ we fix

$$\Psi(x) = x^{-i\sqrt{b/\tau_H}}(1+\cdots). \tag{69}$$

This then implies that in the far region the terms in (67) are

$$\psi_1(\omega, k) = \tau_H^{1-\Delta_{\text{CE}}-i\sqrt{\frac{b}{\tau_H}}} \frac{\Gamma(2\Delta_{\text{CE}}-1)\,\Gamma\left(1-2i\sqrt{\frac{b}{\tau_H}}\right)}{\Gamma\left(\Delta_{\text{CE}}-i\sqrt{\frac{b}{\tau_H}}-i\sqrt{\frac{a}{\tau_H}}\right)\Gamma\left(\Delta_{\text{CE}}-i\sqrt{\frac{b}{\tau_H}}+i\sqrt{\frac{a}{\tau_H}}\right)},$$

$$\psi_2(\omega, k) = \tau_H^{\Delta_{\text{CE}}-i\sqrt{\frac{b}{\tau_H}}} \frac{\Gamma(1-2\Delta_{\text{CE}})\,\Gamma\left(1-2i\sqrt{\frac{b}{\tau_H}}\right)}{\Gamma\left(1-\Delta_{\text{CE}}-i\sqrt{\frac{b}{\tau_H}}-i\sqrt{\frac{a}{\tau_H}}\right)\Gamma\left(1-\Delta_{\text{CE}}-i\sqrt{\frac{b}{\tau_H}}+i\sqrt{\frac{a}{\tau_H}}\right)}. \tag{70}$$

From this we can read off the two-point function to be

$$G_{\text{CE}}(\omega, k) = \frac{\psi_2(\omega, k)}{\psi_1(\omega, k)}. \tag{71}$$

Up to an overall normalization, the dependence on gamma functions in (71) agrees with a WCFT retarted Green's function reported in [26]. Here we are selecting a simple normalization of the correlator, that we will be consistent between the CE and QE WBH. It is worth remarking that this not the standard normalization used for free fields in AdS$_{d+1}$, see for example [79,80], nor the equivalent derivation done in [72].

The next step is to report on the low-temperature and low-frequency behaviour of (71). Implementing the decoupling limit (52) on the frequency and momenta we find,

$$k_{\text{ir}} = k, \qquad \epsilon\, \omega_{\text{ir}} = 2\frac{\ell_2}{\ell^2}R_0\left(\omega - \frac{k\ell}{R_0}\right), \tag{72}$$

where $(\omega_{\text{ir}}, k_{\text{ir}})$ are conjugate to $(\tau, \psi)$. Note that in the limit $\epsilon \to 0$ one holds $\omega_{\text{ir}}$ and $k_{\text{ir}}$ fixed.[13] The coefficients in (66) then become

$$a(\omega_{\text{ir}}, k_{\text{ir}}) = \frac{\tau_H}{4\eth^2}\ell_2^2\left(\omega_{\text{ir}} - 2\nu\eth\frac{\ell_2}{R_0}k_{\text{ir}}\right)^2,$$

$$b(\omega_{\text{ir}}, k_{\text{ir}}) = \frac{\tau_H}{4\eth^2}\ell_2^2\left(\omega_{\text{ir}} + 2\nu\eth\frac{\ell_2}{R_0}k_{\text{ir}}\right)^2, \tag{73}$$

$$\Delta_{\text{CE}} = \frac{1}{2} + \frac{1}{2}\sqrt{1 + 4\ell_2^2 m^2 + 12\frac{\ell_2^4}{R_0^2\ell^2}(1-\nu^2)k_{\text{ir}}^2},$$

---

[12]It is important to stress that this is due to a local $sl(2,\mathbb{R})$ factor present in all of these spaces. In higher dimensions this is no longer true, and the matching procedure is more delicate.

[13]It is useful again to compare with [72]. There the authors take $R_0 \gg 1$ and this suppresses the momentum dependence in (72). To keep the discussion more general, we will take $R_0$ large but fixed. In this context, we are following [78], which is a near to superradiance limit.

where we are reporting on their leading behaviour in the limit $\epsilon \to 0$. It is important to mention that these are exactly the coefficients one would obtain in (66) when the Klein-Gordon operator is evaluated on the near-horizon background (54); this is part of matching procedure, which works easily in this geometry. With this, the two-point function in the near-AdS$_2$ regime is

$$G_{\text{CE}}(\omega_{\text{ir}}, k_{\text{ir}}) = \tau_H^{2\Delta_{\text{CE}}-1} \frac{\Gamma(1-2\Delta_{\text{CE}})}{\Gamma(2\Delta_{\text{CE}}-1)} \frac{\Gamma\left(\Delta_{\text{CE}} - i\frac{\ell_2}{\tilde{\delta}}\omega_{\text{ir}}\right)\Gamma\left(\Delta_{\text{CE}} - i2\nu\frac{\ell_2^2}{R_0}k_{\text{ir}}\right)}{\Gamma\left(1-\Delta_{\text{CE}} - i\frac{\ell_2}{\tilde{\delta}}\omega_{\text{ir}}\right)\Gamma\left(1-\Delta_{\text{CE}} - i2\nu\frac{\ell_2^2}{R_0}k_{\text{ir}}\right)}. \tag{74}$$

At $k_{\text{ir}} = 0$, or alternatively $R_0 \gg 1$, this expression reduces to

$$G_{\text{CE}}(\omega_{\text{ir}}) = \tau_H^{2\Delta_{\text{CE}}-1} \frac{\Gamma(1-2\Delta_{\text{CE}})\Gamma(\Delta_{\text{CE}})}{\Gamma(2\Delta_{\text{CE}}-1)\Gamma(1-\Delta_{\text{CE}})} \frac{\Gamma\left(\Delta_{\text{CE}} - i\frac{\ell_2}{\tilde{\delta}}\omega_{\text{ir}}\right)}{\Gamma\left(1-\Delta_{\text{CE}} - i\frac{\ell_2}{\tilde{\delta}}\omega_{\text{ir}}\right)}$$
$$\sim \left(8\pi\frac{\ell_2^2}{\ell^2}\frac{R_0}{r_0}\frac{\ell}{\beta^{\text{CE}}}\right)^{2\Delta_{\text{CE}}-1} G_{\text{AdS}_2}(\omega_{\text{ir}}), \tag{75}$$

where now $\Delta_{\text{CE}}$ is independent of the momentum; $R_0$ was defined in (52) and the temperature $\beta^{\text{CE}} = 1/T^{\text{CE}}$ and level are defined in (14) and (18) respectively. Since we have been zooming into extremality to derive (75) it only makes sense as long as $\nu \neq 1$, as we also remark below (44). $G_{\text{AdS}_2}(\omega_{\text{ir}})$ is the greybody factor one would obtain in thermal AdS$_2$. In the last line we are being cavalier about the normalization of $G_{\text{AdS}_2}$, but any ambiguity here is frequency independent. The relation (75) is in accordance with similar results obtained for BTZ in [72].

## 4 Near-extremal warped black holes: Quadratic ensemble

In this section we turn to the warped black hole metric in the quadratic ensemble (QE), described in Sec. 2.2. The analysis mirrors the canonical ensemble (CE) in Sec. 3. We perform a similar near-horizon analysis. The contrast between CE and QE is delegated to Sec. 6.

### 4.1 Thermodynamics

In the following we will introduce the concepts of "extremality" and "near-extremality" for the QE black hole from a thermodynamic perspective. Our analysis follows the structure in the CE ensemble—see Sec. 3.1.

The extremal black hole is defined as the solution (23) for which

$$\textbf{Extremality:} \quad r_+ = r_- \equiv r_0. \tag{76}$$

At extremality, the values of the potentials (30) is

$$T^{\text{QE}}\bigg|_{r_\pm = r_0} = 0,$$
$$\Omega^{\text{QE}}\bigg|_{r_\pm = r_0} = -1, \tag{77}$$

while the charges (27) become

$$M_{\text{ext}}^{\text{QE}} \equiv M^{\text{QE}}\bigg|_{r_\pm = r_0} = \frac{r_0^2}{6G_3 L}\sqrt{1-2H^2},$$
$$J_{\text{ext}}^{\text{QE}} \equiv J^{\text{QE}}\bigg|_{r_\pm = r_0} = -\frac{r_0^2}{6G_3 L}\sqrt{1-2H^2}, \tag{78}$$
$$S_{\text{ext}}^{\text{QE}} \equiv S^{\text{QE}}\bigg|_{r_\pm = r_0} = \frac{\pi r_0}{3G_3}\sqrt{1-2H^2}.$$

Near extremality is a small deviation from extremality leading to a non-vanishing temperature, while keeping the angular momentum $J_{\text{ext}}^{\text{QE}}$ fixed. This can be achieved by modifying (76) as

$$\textbf{Near-extremality:} \quad r_+ = r_0 + \epsilon \, \mathfrak{d}, \quad r_- = r_0 - \epsilon \, \mathfrak{d}. \tag{79}$$

Similar to the CE black hole, here $\epsilon \ll 1$, that is a small parameter that introduces the small deviation from extremality; $\mathfrak{d}$ is a parameter that will remain fixed as one takes $\epsilon \to 0$ and was chosen to keep $J_{\text{ext}}^{\text{QE}}$ fixed at leading order in $\epsilon$. The leading order response in $\epsilon$ of the temperature is linear, and reads

$$T^{\text{QE}} = \frac{2}{L\pi} \epsilon \, \mathfrak{d} + \mathcal{O}(\epsilon^2). \tag{80}$$

The mass on the other hand increases quadratically

$$\Delta E^{\text{QE}} = M^{\text{QE}} - M_{\text{ext}}^{\text{QE}} = \frac{(T^{\text{QE}})^2}{M_{\text{gap}}^{\text{CE}}} + \mathcal{O}(\epsilon^3), \tag{81}$$

where the commonly coined mass gap is given by

$$M_{\text{gap}}^{\text{QE}} \equiv \frac{6G_3\sqrt{1-2H^2}}{\pi^2 L(1-H^2)} = \frac{12}{\pi^2 c}. \tag{82}$$

Here we have used (34) to relate the mass gap to the central charge associated to the asymptotic symmetry group in the quadratic ensemble. It also follows that the entropy responds linearly in temperature as we deviate from extremality with the slope inversely proportional to the mass gap:

$$S^{\text{QE}} = S_{\text{ext}}^{\text{QE}} + 2\frac{T^{\text{QE}}}{M_{\text{gap}}^{\text{QE}}} + \mathcal{O}(\epsilon^2). \tag{83}$$

## 4.2 Decoupling limit

In this portion we will report on the geometrical effects of the near-extremal limit. Again, the analysis is very analogous to Sec. 3.2, therefore we will only present the main equations and minimal commentary.

To zoom into the horizon in the near-extremal regime, we introduce a new coordinate system $(\rho, \tau, \psi)$ which redefines the coordinates $(r, t, \theta)$ used in (23). The transformation reads

$$
\begin{aligned}
r &= r_0 + \epsilon \left( e^{\rho/\ell_2} + \frac{\mathfrak{d}^2}{4} e^{-\rho/\ell_2} \right), \\
t &= \ell_2 \frac{\tau}{\epsilon}, \\
\varphi &= \psi + \frac{\ell_2}{L}\frac{\tau}{\epsilon},
\end{aligned}
\tag{84}
$$

where $\epsilon$ and $\mathfrak{d}$ are defined around (79). Here the AdS$_2$ radius is also given by (53), and it is interesting to note that in the nomenclature of QE, we have

$$\ell_2 \equiv \frac{L}{2}, \tag{85}$$

where $L$ is the effective AdS$_3$ radius, defined in (25). The near-horizon region is defined by using (84) on (23) and taking the limit $\epsilon \to 0$, while keeping all other parameters fixed. The

resulting line element is

$$ds_{\text{QE}}^2 = d\rho^2 - e^{2\rho/\ell_2}\left(1 - \frac{\partial^2}{4}e^{-2\rho/\ell_2}\right)^2 d\tau^2$$
$$+ R_0^2\left(d\psi + \frac{1}{R_0}\sqrt{1-2H^2}\,e^{\rho/\ell_2}\left(1 + \frac{\partial^2}{4}e^{-2\rho/\ell_2}\right)d\tau\right)^2 + \mathcal{O}(\epsilon), \tag{86}$$

where we have defined

$$R_0 \equiv R(r_0)\Big|_{r_\pm = r_0} = r_0\sqrt{1-2H^2}, \tag{87}$$

which is equivalent to (53) due to the ties in (41). This locally AdS$_2$ solution is exactly the same as (54).

As we remarked for the CE black hole, it will also be important to quantify how the solution responds to the first order away from extremality. We again parametrize the first response in $\epsilon$ as

$$ds_{\text{QE}}^2 = \left(\bar{g}_{\mu\nu} + \epsilon\, h_{\mu\nu}\right)dx^\mu dx^\nu + \left(R_0^2 + \epsilon\mathscr{Y}\right)\left(d\psi + (\bar{A}_\mu + \epsilon\mathscr{A}_\mu)dx^\mu\right)^2 + \cdots. \tag{88}$$

For the QE black hole the responses are given by

$$\mathscr{Y} = 2R_0\sqrt{1-2H^2}\,e^{\rho/\ell_2}\left(1 + \frac{\partial^2}{4}e^{-2\rho/\ell_2}\right),$$
$$\mathscr{A} = \frac{1}{2R_0^2}(3-2H^2)\,e^{2\rho/\ell_2}\left(1 + \frac{\partial^4}{16}e^{-4\rho/\ell_2}\right)d\tau + \frac{1}{4R_0^2}(1-6H^2)\,d\tau,$$
$$h_{\tau\tau} = \frac{\sqrt{1-2H^2}}{R_0}\,e^{3\rho/\ell_2}\left(1 - \frac{\partial^2}{4}e^{-2\rho/\ell_2}\right)^2\left(1 + \frac{\partial^2}{4}e^{-2\rho/\ell_2}\right), \tag{89}$$
$$h_{\rho\rho} = \frac{\sqrt{1-2H^2}}{R_0}\,e^{\rho/\ell_2}\left(1 + \frac{\partial^2}{4}e^{-2\rho/\ell_2}\right), \quad h_{\tau\rho} = 0.$$

An obvious difference relative to (56) is that we are not preserving the radial gauge: $h_{\rho\rho}$ is non-zero. Conceptually this is not a problem. It can be restored by doing a re-definition of the radial coordinate.

### 4.3 Two-point function

Following the analysis in Sec. 3.3, in this section we discuss the Klein-Gordon equation of a massive scalar field when the background is given by the near-extremal QE black hole (23). More explicitly, we will solve

$$\nabla^2 \Phi(t, r, \varphi) = m^2 \Phi(t, r, \varphi), \tag{90}$$

and report on the behaviour at low energies. To solve this equation, we use a separable ansatz for $\Phi$ to further decompose it into Fourier modes; we will write

$$\Phi(t, r, \varphi) = \sum_k \int d\omega\, e^{-i\frac{\omega}{L}t + ik\varphi}\Psi(r). \tag{91}$$

Here, we are using $L = 2\ell_2$ as the unit of time since it is the parameter that naturally enters in the QE ensemble. Notice that we are abusing the notation here to avoid clutter: $(\omega, k)$ in (91) are not equal to those used in (58), since the notion of time is different for each geometry—see (41). Using (91), we obtain the wave equation

$$\frac{1}{r}\frac{\partial}{\partial r}\left((r^2 - r_+^2)(r^2 - r_-^2)\frac{1}{r}\frac{\partial}{\partial r}\right)\Psi(r)$$
$$+ \left(\frac{1}{N^\varphi_{\text{QE}}(r)^2}(\omega + N^\varphi(r)Lk)^2 - \frac{L^2 k^2}{R_{\text{QE}}(r)^2}\right)\Psi(r) = L^2 m^2 \Psi(r). \tag{92}$$

The functions $N_{\text{QE}}(r)^2$, $R_{\text{QE}}(r)^2$ and $N^\varphi(r)$ are defined in (24). To analyse the greybody factors, we will introduce very similar variables as in Sec. 3.3. We define

$$x \equiv \frac{r^2 - r_+^2}{r_+^2}, \quad \tau_H = \frac{r_+^2 - r_-^2}{r_+^2},$$  (93)

where again, to avoid clutter, the notation is abused relative to (60). Notice that for the QE ensemble, we have $\tau_H = \frac{2\pi L}{r_+} T^{\text{QE}}$, where $T^{\text{QE}}$ is the temperature of the QE black hole. With these definitions, (92) becomes

$$\frac{\partial}{\partial x}\left(x(x+\tau_H)\frac{\partial}{\partial x}\right)\Psi(x) + \frac{L^2}{4\tau_H r_+^4}\frac{(r_+\omega - r_- k)^2}{x}\Psi(x) - \frac{L^2}{4\tau_H r_+^4}\frac{(r_-\omega - r_+ k)^2}{x+\tau_H}\Psi(x)$$
$$= \frac{L^2}{4}m_{\text{QE}}^2\Psi(r),$$  (94)

with

$$m_{\text{QE}}^2 \equiv m^2 + \frac{2H^2}{(1-2H^2)}\frac{(\omega+k)^2}{(r_++r_-)^2}.$$  (95)

There are a few features that are worth highlighting. First the left-hand side of (94) is the wave equation of BTZ, which appears in (26). In that context the effects of the warping appear all in the right-hand side as a distortion of the mass of the probe. Note that this shift in the mass vanishes when $H^2 = 0$, i.e., in the limiting BTZ case.

Another key feature is that (92) has the same structure as (66), and hence it is straightforward to report on the grebody factors. The steps between (66)-(71) are exactly the same, and hence we have

$$G_{\text{QE}}(\omega, k) = \frac{\psi_2(\omega, k)}{\psi_1(\omega, k)},$$  (96)

with

$$\psi_1(\omega, k) = \tau_H^{1-\Delta_{\text{QE}}-i\sqrt{\frac{b}{\tau_H}}} \frac{\Gamma\left(2\Delta_{\text{QE}}-1\right)\Gamma\left(1-2i\sqrt{\frac{b}{\tau_H}}\right)}{\Gamma\left(\Delta_{\text{QE}}-i\sqrt{\frac{b}{\tau_H}}-i\sqrt{\frac{a}{\tau_H}}\right)\Gamma\left(\Delta_{\text{QE}}-i\sqrt{\frac{b}{\tau_H}}+i\sqrt{\frac{a}{\tau_H}}\right)},$$
$$\psi_2(\omega, k) = \tau_H^{\Delta_{\text{QE}}-i\sqrt{\frac{b}{\tau_H}}} \frac{\Gamma\left(1-2\Delta_{\text{QE}}\right)\Gamma\left(1-2i\sqrt{\frac{b}{\tau_H}}\right)}{\Gamma\left(1-\Delta_{\text{QE}}-i\sqrt{\frac{b}{\tau_H}}-i\sqrt{\frac{a}{\tau_H}}\right)\Gamma\left(1-\Delta_{\text{QE}}-i\sqrt{\frac{b}{\tau_H}}+i\sqrt{\frac{a}{\tau_H}}\right)},$$  (97)

and $a(\omega, k)$ and $b(\omega, k)$ are the coefficients of $(x+\tau_H)^{-1}$ and $x^{-1}$ in (94), respectively. We have also introduced

$$\Delta_{\text{QE}} \equiv \frac{1}{2} + \frac{1}{2}\sqrt{1 + L^2 m_{\text{QE}}^2},$$  (98)

which is a frequency and momentum dependent "conformal dimension".

Next, we can take the near-extremal limit. From (84), we have that

$$k = k_{\text{ir}}, \quad \epsilon\omega_{\text{ir}} = \frac{1}{2}(\omega - k),$$  (99)

in combination with (79). We will taking the limit $\epsilon \to 0$ while keeping $\omega_{\text{ir}}$ and $k_{\text{ir}}$ fixed. In this limit we have

$$a(\omega_{\text{ir}}, k_{\text{ir}}) = \frac{\tau_H}{16\partial^2}L^2\left(\omega_{\text{ir}} - \frac{\partial}{r_0}k_{\text{ir}}\right)^2,$$
$$b(\omega_{\text{ir}}, k_{\text{ir}}) = \frac{\tau_H}{16\partial^2}L^2\left(\omega_{\text{ir}} + \frac{\partial}{r_0}k_{\text{ir}}\right)^2,$$  (100)
$$\Delta_{\text{QE}} = \frac{1}{2} + \frac{1}{2}\sqrt{1 + L^2 m^2 + \frac{2H^2}{(1-2H^2)}\frac{L^2}{r_0^2}k_{\text{ir}}^2}.$$

It is important to notice that $a/\tau_H$, $b/\tau_H$ and $\Delta_{\text{QE}}$ here exactly agree with those in the canonical ensemble in (73); a useful identity to check this is

$$r_0^2 = \frac{\nu^2 + 3}{4\nu^2} R_0^2 = \frac{\ell^2}{\ell_2^2} \frac{R_0^2}{4\nu^2}, \tag{101}$$

which arises from (41). This is also what we expect, since the near-horizon geometry in the canonical ensemble (54) is the same as the one in the quadratic ensemble (86). And hence our definitions of $\omega_{\text{ir}}$ and $k_{\text{ir}}$ are the same in both cases.

Finally, we report on the greybody factor when $k_{\text{ir}} = 0$, which reads

$$
\begin{aligned}
G_{\text{QE}}(\omega_{\text{ir}}) &= \tau_H^{2\Delta_{\text{QE}}-1} \frac{\Gamma(1-2\Delta_{\text{QE}})\Gamma(\Delta_{\text{QE}})}{\Gamma(2\Delta_{\text{QE}}-1)\Gamma(1-\Delta_{\text{QE}})} \frac{\Gamma\left(\Delta_{\text{QE}} - i\frac{\ell_2}{\partial}\omega_{\text{ir}}\right)}{\Gamma\left(1-\Delta_{\text{QE}} - i\frac{\ell_2}{\partial}\omega_{\text{ir}}\right)} \\
&\sim \left(\frac{2\pi L}{r_0} \frac{1}{\beta^{\text{QE}}}\right)^{2\Delta_{\text{QE}}-1} G_{\text{AdS}_2}(\omega_{\text{ir}}).
\end{aligned}
\tag{102}
$$

This follows in a straightforward way from the derivations in the canonical ensemble around (75). The temperature $\beta^{\text{QE}} \equiv 1/T^{\text{QE}}$ in (102) is defined in (30).

# 5 A two-dimensional perspective of warped black holes

Until now we have explored near-extremal properties of WBH starting from the non-extremal solutions. That is, we have captured the near zero temperature behaviour by taking appropriate limits of the finite temperature black hole. In this section we will set up the stage to reverse this logic: we want to capture the near-extremal behaviour by deforming away from the extremal, zero temperature black hole.

A systematic way to proceed is to view these black holes from a two-dimensional perspective. More explicitly, we will perform a dimensional reduction along a compact direction. The way we will decompose our three-dimensional spacetime is as follows,

$$ds_3^2 = g_{MN}\mathrm{d}x^M\mathrm{d}x^N = g_{\mu\nu}\mathrm{d}x^\mu\mathrm{d}x^\nu + e^{-2\phi}\left(\mathrm{d}z + A_\mu\mathrm{d}x^\mu\right)^2, \tag{103}$$

where the Greek indices run along the two-dimensional directions, $\mu, \nu = 0, 1$, and $z$ is a compact direction with $z \sim z + 2\pi$. We will be trading the three-dimensional metric $g_{MN}$ for the two-dimensional variables: a two-dimensional metric $g_{\mu\nu}$, a gauge field $A_\mu$ and a dilaton field $\phi$. The working assumption is that all the variables are independent of $z$, which is a truncation of the three-dimensional theory, but it will suffice to describe the near-extremal system.

The effects of this dimensional reduction on the three-dimensional action (1) are known [81,82], and we will follow the conventions in [71]. The resulting two-dimensional theory is

$$I_{\text{2D}} = I_{\text{EMD}} + I_{\text{rCS}}. \tag{104}$$

$I_{\text{EMD}}$ is a two-dimensional Einstein-Maxwell-Dilaton theory whose couplings are dictated by the dimensional reduction of the Einstein-Hilbert term in (1); it reads

$$I_{\text{EMD}} = \frac{1}{8G_3} \int \mathrm{d}^2x \sqrt{-g}\, e^{-\phi}\left(\mathscr{R} + \frac{2}{\ell^2} - \frac{1}{4}e^{-2\phi} F_{\mu\nu}F^{\mu\nu}\right). \tag{105}$$

In this expression, $\mathscr{R}$ is the two-dimensional Ricci scalar associated to the metric $g_{\mu\nu}$, and the field strength is given by $F_{\mu\nu} = \partial_\mu A_\nu - \partial_\nu A_\mu$. $I_{\text{rCS}}$, the 'reduced Chern-Simons' term, contains

the information from the dynamics of the gravitational Chern-Simons term in (1), and it is given by

$$I_{\text{rCS}} = \frac{1}{32 G_3 \mu} \int \mathrm{d}^2 x \, e^{-2\phi} \epsilon^{\mu\nu} \left( F_{\mu\nu} \mathscr{R} + F_{\mu\rho} F^{\rho\sigma} F_{\sigma\nu} e^{-2\phi} - 2 F_{\mu\nu} \nabla^2 \phi \right) . \tag{106}$$

Here $\epsilon^{\mu\nu}$ is the epsilon symbol, where $\epsilon^{01} = 1$, and $\nabla_\mu$ is the covariant derivative with respect to the two-dimensional metric $g_{\mu\nu}$.

The equations of motion one obtains from (104) are given by

$$\epsilon^{\alpha\beta} \partial_\beta \left( e^{-3\phi} f + \frac{1}{2\mu} e^{-2\phi} \left( \mathscr{R} + 3 e^{-2\phi} f^2 - 2 \nabla^2 \phi \right) \right) = 0 ,$$

$$e^{-\phi} \left( \mathscr{R} + \frac{2}{\ell^2} + \frac{3}{2} e^{-2\phi} f^2 \right) + \frac{1}{\mu} e^{-2\phi} f \left( \mathscr{R} + 2 e^{-2\phi} f^2 - 2 \nabla^2 \phi \right) + \frac{1}{\mu} \nabla^2 \left( e^{-2\phi} f \right) = 0 ,$$

$$g_{\alpha\beta} \left( \nabla^2 e^{-\phi} - \frac{1}{\ell^2} e^{-\phi} + \frac{1}{4} e^{-3\phi} f^2 \right) - \nabla_\alpha \nabla_\beta e^{-\phi} \tag{107}$$

$$+ \frac{1}{2\mu} \left( (\nabla_\alpha e^{-2\phi} f) \nabla_\beta \phi + (\nabla_\beta e^{-2\phi} f) \nabla_\alpha \phi - \nabla_\alpha \nabla_\beta (e^{-2\phi} f) \right)$$

$$+ \frac{1}{2\mu} g_{\alpha\beta} \left( \frac{1}{2} e^{-2\phi} f \mathscr{R} - e^{-2\phi} f \nabla^2 \phi - \nabla_\mu (e^{-2\phi} f) \nabla^\mu \phi + \nabla^2 (e^{-2\phi} f) + e^{-4\phi} f^3 \right) = 0 ,$$

where we have introduce the auxiliary scalar

$$f \equiv \frac{1}{2\sqrt{-g}} \epsilon^{\alpha\beta} F_{\alpha\beta} . \tag{108}$$

As stressed in [71,81,82], the action (104) is a consistent truncation of the three-dimensional theory. That is, all solutions to the equations of motion (107), when uplifted via (103), are solutions to (3).

The solutions that will serve as our base in the subsequent analysis are those that have a *constant dilaton* background. As shown in [71], the equations of motion (107) admit two branches of solutions when we set $\phi(x) = \phi_0$ constant. The first branch is characterised by being a solution that is independent of the TMG coupling $\mu$; that is, it is determined by the equations that arise from the EMD action, and hence it automatically satisfies also (107). The Ricci scalar and auxiliary scalar $f$ (108) are

$$\textbf{EMD Branch}: \quad \mathscr{R}_0 = -\frac{8}{\ell^2} , \qquad f_0^2 = \frac{4}{\ell^2} e^{2\phi_0} . \tag{109}$$

The second branch is a solution that relies on a balance between the EMD and rCS contributions to (107), and hence is intrinsic to the TMG dynamics. The corresponding Ricci scalar and auxiliary scalar read

$$\textbf{TMG Branch}: \quad \mathscr{R}_0 = -\frac{6}{\ell^2} - \frac{2\mu^2}{9} , \qquad f_0 = -\frac{2\mu}{3} e^{\phi_0} . \tag{110}$$

For both branches the subscript "0" simply denotes the background values when we set the dilaton to be a constant. For both of these branches, the Ricci scalar is constant, and negative, indicating the two-dimensional metric is locally AdS$_2$, as expected. From here we will identify the AdS$_2$ radius via the relation

$$\mathscr{R}_0 = -\frac{2}{\ell_2^2} . \tag{111}$$

The EMD branch is the appropriate solution to describe the near-horizon geometry of the extremal BTZ black hole, as is explained in [71]. The TMG branch is the appropriate solution to describe the near-horizon of extremal WBH, both in the canonical and quadratic ensemble. We will show this explicitly below, and at this stage we just remark that the radius of the AdS$_2$ space in (110) agrees with (53) as it should.

## 5.1 Near-AdS$_2$: Linear response

We are interested in small fluctuations about our AdS$_2$ background solution (110). We will parametrize these fluctuations as follows,

$$
\begin{aligned}
e^{-2\phi} &= e^{-2\phi_0} + \mathcal{Y}\,, \\
f &= f_0 + \mathcal{F}\,, \\
g_{\alpha\beta} &= \overline{g}_{\alpha\beta} + h_{\alpha\beta}\,.
\end{aligned}
\tag{112}
$$

Here the fields $(\phi_0, f_0, \overline{g}_{\alpha\beta})$ will correspond to the TMG branch in (110); in particular $\overline{g}_{\alpha\beta}$ is a locally AdS$_2$ metric whose curvature is given by $\mathcal{R}_0$. In the following we will describe the dynamics of the fluctuations $(\mathcal{Y}, \mathcal{F}, h_{\alpha\beta})$, whose support is on the two-dimensional coordinates $x^\mu$, at the linearized level. We will describe the equations of motion and the effective action that describes this leading order response.

By expanding the equations of motion (107) around (112), the linearized equations of motion read

$$
e^{2\phi_0}\left(\overline{\nabla}^2 + \frac{4\mu^2}{9} - \frac{6}{\ell^2}\right)\mathcal{Y} - 2\mu e^{-\phi_0}\mathcal{F} + \delta\mathcal{R} = 0\,,
$$

$$
4e^{2\phi_0}\left(\overline{\nabla}^2 + \mu^2 - \frac{3}{4\ell^2}\right)\mathcal{Y} - \frac{3}{\mu}e^{-\phi_0}\left(\overline{\nabla}^2 + \frac{4}{9}\mu^2 - \frac{6}{\ell^2}\right)\mathcal{F} - \delta\mathcal{R} = 0\,,
$$

$$
e^{2\phi_0}\left(\overline{\nabla}_\alpha\overline{\nabla}_\beta + \overline{g}_{\alpha\beta}\left\{\frac{5\mu^2}{9} - \frac{3}{\ell^2}\right\}\right)\mathcal{Y} + \frac{3}{\mu}e^{-\phi_0}\left(\overline{\nabla}_\alpha\overline{\nabla}_\beta - \overline{g}_{\alpha\beta}\left\{\overline{\nabla}^2 + \frac{5\mu^2}{9} - \frac{3}{\ell^2}\right\}\right)\mathcal{F}
$$
$$
+ \overline{g}_{\alpha\beta}\,\delta\mathcal{R} = 0\,,
\tag{113}
$$

where $\overline{\nabla}$ stands for the covariant derivative with respect to the background metric $\overline{g}_{\alpha\beta}$. The fluctuation of the Ricci scalar, $\delta\mathcal{R}$, which contains the terms depending on $h_{\alpha\beta}$, is

$$
\delta\mathcal{R} = \overline{\nabla}^\alpha\overline{\nabla}^\beta h_{\alpha\beta} - \overline{\nabla}^2 h^\alpha_{\ \alpha} + \frac{1}{\ell_2^2}h^\alpha_{\ \alpha}\,.
\tag{114}
$$

The equations in (113) couple our three fluctuations, but it is possible to decouple the system systematically. First, we can use the first two equations in (113) to solve for $\delta\mathcal{R}$ and replace in the third equation. This gives the following equation

$$
\overline{\nabla}_\alpha\overline{\nabla}_\beta\Phi(x) - \overline{g}_{\alpha\beta}\overline{\nabla}^2\Phi(x) + \frac{1}{\ell_2^2}\overline{g}_{\alpha\beta}\Phi(x) = 0\,,
\tag{115}
$$

where

$$
\Phi(x) \equiv 3\mathcal{F}(x) + e^{3\phi_0}\mu\mathcal{Y}(x)\,.
\tag{116}
$$

This is the characteristic equation for Jackiw-Teitelboim (JT) gravity [40, 41]. For this reason we will refer to $\Phi(x)$ as the "dilaton." However, in sharp contrast to other instances of JT gravity, $\Phi$ does not parametrize the size of the black hole horizon ($\mathcal{Y}$ plays that role). With this, we can rewrite (113) in terms of $(\Phi(x), \mathcal{F}, h_{\mu\nu})$; the first equation is (115) and the remaining two are

$$
\left(\overline{\nabla}^2 - \frac{1}{\ell_2^2} + \frac{4\mu^2}{9}\right)\mathcal{F} - \frac{1}{3}\left(\frac{1}{\ell_2^2} + \frac{2}{9}\mu^2\right)\Phi = 0\,,
$$
$$
\delta\mathcal{R} + \frac{e^{-\phi_0}}{\mu}\left(\left(\frac{8\mu^2}{3} - 3\ell_2^{-2}\right)\mathcal{F} + \left(-\frac{4\mu^2}{9} + \ell_2^{-2}\right)\Phi\right) = 0\,.
\tag{117}
$$

It is worth analysing the physical interpretation of these fluctuations. From (115), we see that the conformal dimension of $\Phi$ is $\Delta_\Phi = 2$. The solution to the first equation in (117)

contains a homogeneous and inhomegenous part, i.e.,

$$\mathscr{F}(x) = \mathscr{F}_{\text{hom}}(x) + \frac{1}{3}\left(\frac{9 + 2\ell_2^2\mu^2}{9 + 4\ell_2^2\mu^2}\right)\Phi(x), \tag{118}$$

with the homogenous part satisfying

$$\left(\overline{\nabla}^2 - \frac{1}{\ell_2^2} + \frac{4\mu^2}{9}\right)\mathscr{F}_{\text{hom}}(x) = 0. \tag{119}$$

$\mathscr{F}_{\text{hom}}$ is an independent degree of freedom, and it can be tracked to the extra degree of freedom due to the appearance of a massive graviton that is characteristic to TMG. It is useful to further interpret this field in the usual AdS/CFT dictionary. From (119) we can relate the mass of the field to its conformal dimension; this gives

$$\Delta_{\mathscr{F}}(\Delta_{\mathscr{F}} - 1) = 1 - \frac{4}{9}\mu^2\ell_2^2 = \frac{3(1 - \nu^2)}{3 + \nu^2}, \tag{120}$$

where $\nu = \frac{\mu\ell}{3}$ as before, and we used (53). The solutions are

$$\Delta_{\mathscr{F}}^{\pm} = \frac{1}{2}\left(1 \pm \sqrt{\frac{15 - 11\nu^2}{3 + \nu^2}}\right). \tag{121}$$

As we saw in Sec. 2, our WBH solutions have the restriction that $\nu^2 \geq 1$, making the mass squared negative. We also have the Breitenlohner-Freedman (BF) bound [83]: this restricts $\nu^2 \leq \frac{15}{11}$ such that $\Delta_{\mathscr{F}} \geq 0$. Therefore, we have a linear stable mode when

$$1 \leq \nu^2 \leq \frac{15}{11} : \qquad \frac{1}{2} \leq \Delta_{\mathscr{F}}^+ \leq 1, \qquad 0 \leq \Delta_{\mathscr{F}}^- \leq \frac{1}{2}. \tag{122}$$

Altogether, this makes $\mathscr{F}_{\text{hom}}(x)$ a relevant perturbation, and being marginal when $\nu^2 = 1$, around the AdS$_2$ background. This mode, and its non-trivial bounds, were also found in [42]; an interesting contrast is that here we detected it from an analysis of the IR (AdS$_2$) background rather than from the fluctuations around Warped AdS$_3$.

Finally, it is worth reporting on the effective action that captures the linear response. The equations of motion obtained from

$$\begin{aligned}
I_{\text{eff}} = &\frac{e^{-4\phi_0}}{48G_3}\int d^2x\sqrt{-g}\,\Phi\left(R + \frac{2}{\ell_2^2}\right) \\
&- \frac{9e^{-3\phi_0}}{16\mu G_3}\int d^2x\sqrt{-g}\left(\overline{\nabla}_\mu\mathscr{F}\,\overline{\nabla}^\mu\mathscr{F} + \frac{1}{\ell_2^2}\Delta_{\mathscr{F}}(\Delta_{\mathscr{F}} - 1)\mathscr{F}^2\right) \\
&+ \frac{e^{-3\phi_0}}{48\mu G_3}\int d^2x\sqrt{-g}\left(\frac{1}{3}\left(\mu^2 + \frac{9}{\ell_2^2}\right)\Phi^2 - 4\left(\mu^2 - \frac{3}{\ell_2^2}\right)\mathscr{F}\Phi + 15\overline{\nabla}_\mu\mathscr{F}\,\overline{\nabla}^\mu\Phi\right),
\end{aligned} \tag{123}$$

exactly match (115) and (117) at linear order in the fields. There is an overall factor in $I_{\text{eff}}$ that we fix such that the action here matches the normalization in (104) at the linear level. The first line of (123) is the renown JT action, and for this reason several components of our analysis will agree with the universal properties advocated in [37,38]; the second line contains the kinetic and mass terms for $\mathscr{F}$ (which captures the relevant/marginal operator); and third line of (123) captures the non-trivial interactions among the fields.

For the purposes of capturing dynamics in the near-AdS$_2$ region, the effective action (123) is much simpler to manipulate relative to (105)-(106), since the latter is a higher derivative theory and the former is a two-derivative action. In Sec. 5.2, we will use $I_{\text{eff}}$ to discuss holographic renormalization and thermal properties in the near-AdS$_2$ background.

### 5.1.1 Solutions

In this last portion we will construct explicit solutions to the linear equations (115)-(117); this follows very closely [71,84], which we refer to for further details. It will be convenient to work in a radial gauge, where we introduce coordinates $x^\mu = (\rho, \tau)$ and we set

$$ds^2 = g_{\mu\nu}dx^\mu dx^\nu = \gamma_{\tau\tau}d\tau^2 + d\rho^2, \quad A_\rho = 0. \tag{124}$$

In this gauge, the background AdS$_2$ metric and the gauge field are

$$\bar{g}_{\mu\nu}dx^\mu dx^\nu = \bar{\gamma}_{\tau\tau}d\tau^2 + d\rho^2, \qquad A^0 = A_\tau^0 d\tau, \tag{125}$$

with

$$\begin{aligned}
\bar{\gamma}_{\tau\tau} &= -\left(\alpha(\tau)e^{\rho/\ell_2} + \beta(\tau)e^{-\rho/\ell_2}\right)^2, \\
A_\tau^0 &= \chi(\tau) - f_0\ell_2\left(\alpha(\tau)e^{\rho/\ell_2} - \beta(\tau)e^{-\rho/\ell_2}\right).
\end{aligned} \tag{126}$$

Here $\alpha(\tau)$, $\beta(\tau)$ and $\chi(\tau)$ are arbitrary functions. The constant $f_0$ and the AdS$_2$ radius, $\ell_2$, are given by (110).

With this choice of gauge and background solution, the solution to the JT equation (115) reads

$$\Phi(x) = \lambda(\tau)e^{\rho/\ell_2} + \sigma(\tau)e^{-\rho/\ell_2}, \tag{127}$$

where

$$\begin{aligned}
\sigma(\tau) &= -\frac{\ell_2^2}{4\lambda}\left(\frac{(\partial_\tau\lambda)^2}{\alpha^2} + c_0\right), \\
\beta(\tau) &= -\frac{\ell_2^2}{4}\frac{\alpha}{\partial_\tau\lambda}\partial_\tau\left(\frac{1}{\lambda}\left(\frac{(\partial_\tau\lambda)^2}{\alpha^2} + c_0\right)\right) = \frac{\alpha}{\partial_\tau\lambda}\partial_\tau\sigma.
\end{aligned} \tag{128}$$

Here $c_0$ is an arbitrary constant. Using the standard AdS/CFT terminology, from (127) we interpret $\lambda(\tau)$ as the source and $\sigma(\tau)$ as the vacuum expectation value. In (128) we have chosen to solve for the vacuum expectation values $(\beta, \sigma)$ in terms of the sources $(\alpha, \lambda)$.

Solving the two equations in (117) is straightforward, and we start with the first equation. As described in (118), the field $\mathscr{F}$ has two components. The inhomogeneous solution is

$$\mathscr{F}_{\text{in-hom}}(x) = \frac{\nu^2 + 1}{5\nu^2 + 3}\Phi(x), \tag{129}$$

with $\Phi(x)$ given by (127). The homogeneous solution has the standard behaviours of fields in AdS, whose radial behaviour near the boundary is

$$\mathscr{F}_{\text{hom}}(x) = e^{-\Delta_\mathscr{F}\rho/\ell_2}(f_1(\tau) + \cdots) + e^{(\Delta_\mathscr{F}-1)\rho/\ell_2}(f_2(\tau) + \cdots), \tag{130}$$

with $\Delta_\mathscr{F}$ defined in (120). Since this mode is relevant or marginal, it depends on its quantization conditions if $f_1$, or $f_2$, is the source, or a vacuum expectation value.

The second equation in (117) determines the metric perturbation. Recall that we are working in the radial gauge (124), and hence the only metric perturbation in the game is $h_{\tau\tau}$; the resulting equation is therefore

$$\frac{1}{\bar{\gamma}_{\tau\tau}}\partial_\rho\left(\bar{\gamma}_{\tau\tau}\partial_\rho(\bar{\gamma}^{\tau\tau}h_{\tau\tau})\right) = \frac{e^{-\phi_0}}{\nu\ell}\left((7\nu^2 - 3)\mathscr{F} + (1 - \nu^2)\Phi\right). \tag{131}$$

The solutions to $h_{\tau\tau}$ also split into a homogeneous and inhomogeneous solution. The homogeneous solution is the same as the background solution and can be absorbed in $\bar{\gamma}_{\tau\tau}$. The inhomogeneous solution is determined by the on-shell values we have already determined for $\mathscr{F}$ and $\Phi$. For concreteness, let us take $\mathscr{F} = \mathscr{F}_{\text{in-hom}}(x)$, i.e., turn off the homogeneous solution in (130). With this, the inhomogeneous solution to the metric pertubation reads

$$h_{\tau\tau} = \frac{2\nu\ell}{3}\frac{1}{5\nu^2 + 3}e^{-\phi_0}\left(\bar{\gamma}_{\tau\tau}\Phi - 2\ell_2^2\sqrt{-\bar{\gamma}}\partial_\tau\left(\frac{\partial_\tau\lambda}{\alpha}\right)\right). \tag{132}$$

**Comparison with warped black holes.** In this last portion we compare the solutions described here with the black hole background: the near-AdS$_2$ background in the canonical ensemble Sec. 3.2 and the quadratic ensemble Sec. 4.2. Since they are stationary black holes, they will be matched with static ($\tau$-independent) solutions described in the two-dimensional language used in this section.

The background AdS$_2$ solutions for both black holes are exactly the same, and in terms of the language used in this section it corresponds to

$$e^{-\phi_0} = R_0, \quad \alpha(\tau) = 1, \quad \beta(\tau) = -\frac{\eth^2}{4}, \tag{133}$$

where we used (54) and (86) for the canonical and quadratic solution, respectively. From here we see that $\beta$ is tied to the near-extremal parameter $\eth$.

One small subtlety between the canonical and quadratic ensemble comes from the specific values of the sources in the JT field, $\Phi(x)$. For the canonical black hole we have

$$\lambda_{\mathrm{CE}}(\tau) = \frac{3(3 + 5\nu^2)}{\ell R_0^2} \epsilon, \tag{134}$$

which is obtained by reconstructing $\Phi$ from (116) and the on-shell values in (56).[14] In contrast, the quadratic black hole has

$$\lambda_{\mathrm{QE}}(\tau) = \frac{12(1 - H^2)}{\ell_2 R_0^2} \epsilon, \tag{135}$$

which again uses (116) and the black holes values in (89). To relate (135) to (134) we need to incorporate that the decoupling parameter $\epsilon$ is not the same for the canonical and quadratic ensembles. By relating (52) and (84) via (41), one finds

$$\epsilon_{\mathrm{CE}} = 2\frac{\ell_2}{\ell} \epsilon_{\mathrm{QE}} \quad \Rightarrow \quad \lambda_{\mathrm{QE}}(\tau) = \lambda_{\mathrm{CE}}(\tau). \tag{136}$$

The subleading component of the JT field is simple to read off, and for both cases we have

$$\sigma_{\mathrm{CE,QE}}(\tau) = \frac{\eth^2}{4} \lambda_{\mathrm{CE,QE}}(\tau). \tag{137}$$

Finally, for the massive vector field we find

$$\mathscr{F}_{\mathrm{CE,QE}} = \frac{\nu^2 + 1}{5\nu^2 + 3} \Phi_{\mathrm{CE,QE}}, \tag{138}$$

which shows that, for both black hole backgrounds, the fields on-shell only have the inhomogeneous solution to (117). The remaining fields listed in (56) and (89), i.e., $h_{\mu\nu}$ and $\mathscr{A}_\mu$, will follow from the values listed here, in accordance to the dynamics described in this section.

## 5.2 Boundary analysis

In this final portion, we return to thermodynamic aspects of the warped black holes, but now with the perspective of near-AdS$_2$. We will perform some of the basic computations to read off the entropy of near-AdS$_2$ via a boundary analysis of the system. In a nutshell, we will construct an effective boundary action via the traditional tools of holographic renormalization. From there, we will identify the Schwarzian sector of the theory and report on its contribution to the entropy.

---

[14]Recall that $\epsilon$ is the decoupling parameter used to obtained the near-horizon geometry. In this context, it controls the smallness of $\Phi(x)$ which will use in Sec. 5.2.

The derivations here follow very closely again [71, 84], and conceptually there is no deviation from those references. For that reason we will keep the presentation brief and to the point.

A holographic analysis around the near-AdS$_2$ background requires some care. We are interested in renormalizing the theory when the source of $\Phi(x)$, $\lambda(\tau)$, is turned on; this means we are doing conformal perturbation theory in the presence of irrelevant couplings. In more practical terms for our purposes, we have to specify what are the allowed divergences and the regime of validity of the procedure. Following [84], at a specified cutoff $\rho = \rho_c \to \infty$ we will have

$$\lambda(\tau) e^{\rho_c/\ell_2} \ll 1, \qquad e^{-2\phi_0} \gg \mathscr{Y}. \tag{139}$$

The setup of the variational problem will be standard. Our bulk action is $I_{\text{eff}}$ in (123), and for this action we construct a functional that is well-defined for Dirichlet boundary conditions on the field. With this we will see that the responses of the functional under the variations

$$\delta\gamma_{\tau\tau} = -2\alpha(\tau) e^{2\rho_c/\ell_2} \delta\alpha, \qquad \delta\Phi = e^{\rho_c/\ell_2} \delta\lambda \tag{140}$$

are finite and integrable. Note that we will only turn on the sources for the metric and JT field, $\alpha$ and $\lambda$ respectively; for simplicity we are setting $\mathscr{F}_{\text{hom}} = 0$, which suffices to discuss semiclassical aspects of the thermodynamics of the system that connects to JT gravity. However, it should be stressed that it is of interest to investigate in more details the effects of the massive degree of freedom $\mathscr{F}_{\text{hom}}$. In particular, we expect this mode to lead to instabilities similar to those advocated recently in [85] for higher-dimensional extremal AdS black holes.

The functional that renders a well-defined variational problem for our system is

$$I_{\text{ren}} = I_{\text{eff}} + I_{\text{GH}} + I_{\text{ct}}. \tag{141}$$

The bulk term is given by $I_{\text{eff}}$ in (123). The second term is the usual Gibbons-Hawking term, which in our context reads

$$I_{\text{GH}} = 2\mu e^{\phi_0} \int d\tau \sqrt{-\gamma}\, \Phi K. \tag{142}$$

Here, $K = \partial_\rho \log\sqrt{-\gamma}$ is the extrinsic curvature. The third term in $I_{\text{ren}}$ are local boundary counterterms, whose functionality is to remove divergences in the action (and its variation). The counterterms for our setup are

$$I_{\text{ct}} = -\frac{e^{-2\phi_0}}{24 G_3 \mu \ell_2} \int d\tau \sqrt{-\gamma}\, \Phi + \frac{e^{-3\phi_0} \ell_2}{48 G_3 (9 + 4\mu^2 \ell_2^2)} \int d\tau \sqrt{-\gamma}\, \Phi^2. \tag{143}$$

Although the second term is quadratic in $\Phi$, and hence subleading according to (139), it is needed to render finite the variation of the action with respect to (140).

We can now easily very that the response of $I_{\text{ren}}$ is finite and integrable. First we compute the one-point functions dual to our two sources in (140); this gives

$$\hat{\Pi}_\alpha = \lim_{\rho_c \to \infty} \frac{\delta}{\delta\alpha}(I_{\text{eff}} + I_{\text{GH}} + I_{\text{ct}}) = -\frac{e^{-2\phi_0}}{12 G_3 \ell_2} \sigma(\tau),$$

$$\hat{\Pi}_\lambda = \lim_{\rho_c \to \infty} \frac{\delta}{\delta\lambda}(I_{\text{eff}} + I_{\text{GH}} + I_{\text{ct}}) = -\frac{e^{-2\phi_0}}{12 G_3 \ell_2} \beta(\tau), \tag{144}$$

which is clearly finite. Thus, the variation of the renormalised effective action is

$$\delta I_{\text{ren}} = \int d\tau (\hat{\Pi}_\alpha \delta\alpha + \hat{\Pi}_\lambda \delta\lambda)$$

$$= \frac{\ell_2 e^{-2\phi_0}}{48 G_3 \mu} \int d\tau \left[ \frac{q(\tau)}{\lambda(\tau)} \delta\alpha + \frac{\alpha(\tau)}{\partial_\tau \lambda(\tau)} \partial_\tau \left( \frac{q(\tau)}{\lambda(\tau)} \right) \delta\lambda \right], \tag{145}$$

where we used (128) and defined

$$q(\tau) \equiv \left( \frac{(\partial_\tau \lambda)^2}{\alpha^2} + c_0 \right).$$
(146)

This expression is integrable over the phase space $(\alpha, \lambda)$. After performing this integration over (145), we get

$$I_{\text{ren}} = \frac{\ell_2}{48 G_3 \mu} e^{-2\phi_0} \int d\tau \, \frac{\alpha(\tau)}{\lambda(\tau)} \left( c_0 - \left( \frac{\partial_\tau \lambda(\tau)}{\alpha(\tau)} \right)^2 \right).$$
(147)

Hence we have obtained a finite and well-defined on-shell action for the system at hand.

**Schwarzian effective action.** It is useful to re-cast (147) in a way that the Schwarzian effective action is manifest. To do so, we first make manifest the re-parametrization mode as follows. Take $\alpha = 1$ and $\beta = 0$

$$ds^2 = d\rho^2 - e^{2\rho/\ell_2} d\tau^2.$$
(148)

This is what we coin an "empty" AdS$_2$ background. Next, consider the following diffeomorphism

$$\tau \rightarrow f(\tau) + \frac{\ell_2^2}{2} \frac{f''(\tau)}{e^{2\rho/\ell_2} - \frac{\ell_2^2}{4} \frac{f''(\tau)^2}{f'(\tau)^2}},$$
$$e^{\rho/\ell_2} \rightarrow \frac{e^{-\rho/\ell_2}}{f'(\tau)} \left( e^{2\rho/\ell_2} - \frac{\ell_2^2}{4} \frac{f''(\tau)^2}{f'(\tau)^2} \right),$$
(149)

where $f(\tau)$ is an arbitrary function of time representing boundary time reparametrizations. Under these diffeomorphisms, the line element becomes

$$ds^2 = d\rho^2 - \left( e^{\rho/\ell_2} + \frac{\ell_2}{2} \{f(\tau), \tau\} e^{-\rho/\ell_2} \right)^2 d\tau^2,$$
(150)

where $\{f(\tau), \tau\} = \left( \frac{f''}{f'} \right)' - \frac{1}{2} \left( \frac{f''}{f'} \right)^2$ is the Schwarzian derivative. We see that these diffeomorphisms preserve the radial gauge while ensuring that the asymptotic form of the metric is the same as that of empty AdS$_2$ in (148). So, these are the asymptotic symmetries of AdS$_2$. Comparing with (125), we have

$$\alpha(\tau) = 1, \qquad \beta(\tau) = \frac{\ell_2^2}{2} \{f(\tau), \tau\}.$$
(151)

It is now also simple to re-examine (147) in the view of (151). Substituting for $c_0$ in terms of $\beta$ in (147) by using (128), we get

$$I_{\text{ren}} = \frac{\ell_2}{24 G_3 \mu} e^{-2\phi_0} \int d\tau \left[ \lambda(\tau) \{f(\tau), \tau\} - \frac{\lambda'^2(\tau)}{\lambda(\tau)} \right].$$
(152)

In this expression we used (151); we have also ignored total derivatives, since shortly the time direction will be periodic (in Euclidean signature). This is the well-known Schwarzian effective action that is characteristic of JT gravity [38].

**Near-extremal entropy.** Finally, we turn to extracting some thermodynamic information from (152). For this, we take a static background where all functions are independent of $\tau$. In particular we take[15]

$$\lambda(\tau) = \lambda_0\,, \qquad \beta(\tau) = \beta_0\,. \tag{153}$$

For this configuration, the background $\text{AdS}_2$ solution has a horizon at $e^{2\rho_h/\ell_2} = -\beta_0$. The temperature associated to this horizon is

$$T_{2\text{d}} = \frac{1}{2\pi}\partial_\rho \sqrt{-\bar{\gamma}_{\tau\tau}}\Big|_{\rho=\rho_h} = \frac{1}{\pi\ell_2}\sqrt{|\beta_0|}\,. \tag{154}$$

To extract the entropy, we take an Euclidean approach, for which we evaluate the renormalized action (152) in Euclidean signature. By Wick rotating time, $\tau = i\tau_E$ with $\tau_E \sim \tau_E + T_{2\text{d}}^{-1}$, the action (152) reads

$$I_{\text{ren}}^E = \frac{\ell_2}{48G_3\mu}e^{-2\phi_0}\int_0^{\frac{1}{T_{2\text{d}}}}\mathrm{d}\tau_E\,\lambda_0\,(2\pi T_{2\text{d}})^2 = \frac{\pi^2\ell_2}{12G_3\mu}e^{-2\phi_0}\,\lambda_0\,T_{2\text{d}}\,, \tag{155}$$

where we used (153) and (154). We take the usual relation between the on-shell action and entropy dictated by thermodynamics, which gives

$$S_{2\text{d}} = (1 + T_{2\text{d}}\partial_{T_{2\text{d}}})I_{\text{ren}}^E = \frac{\pi^2\ell_2}{6G_3\mu}e^{-2\phi_0}\,\lambda_0\,T_{2\text{d}}\,. \tag{156}$$

The entropy we have derived here, $S_{2\text{d}}$, is the entropy of the near-$\text{AdS}_2$ background. That is, the entropy as a deviation away from the fixed IR point, which is controlled by $\lambda_0$. It excludes the zero temperature residual entropy, since $I_{\text{eff}}$ and $I_{\text{ren}}$ do not capture that contribution. In our next and final section we will make comparisons with the warped black hole backgrounds in the canonical and quadratic ensemble.

# 6 Comparing perspectives and ensembles

Having done independent analyses in the three previous sections, we now turn to our main task of comparing and contrasting our findings. We will be able to take three different perspectives: in the three-dimensional arena, we will contrast the responses of the WBHs in its two different ensembles; from the two-dimensional dual, where the lamppost is a warped CFT, we will contrast their thermodynamic response; and from the IR perspective of near-$\text{AdS}_2$ dynamics we will disentangle how these come together or apart.

## 6.1 Comparing ensembles

In this first portion, we will scrutinise and contrast the analysis done in Sec. 3 and Sec. 4. This contrast will not involve a holographic component yet, just how the black hole responds in the near-extremal limit from the bulk perspective. To this end, we will be emphasising similarities and differences between the canonical and quadratic ensemble.

**Mass gap.** The response we found for both ensembles is generic: upon moving away from extremality by slightly increasing the temperature, the mass and entropy of the black hole increases quadratically and linearly in the temperature respectively. More explicitly, we have

$$M^\bullet = M_{\text{ext}}^\bullet + \frac{(T^\bullet)^2}{M_{\text{gap}}^\bullet} + \mathcal{O}(\epsilon^3)\,,$$

---

[15]In terms of the time reparametrization mode, we have $f(\tau) = e^{f_0\tau}$, and hence $\beta_0 = -\frac{\ell_2^2}{4}f_0^2$.

$$S^\bullet = S^\bullet_{\text{ext}} + 2\frac{T^\bullet}{M^\bullet_{\text{gap}}} + \mathcal{O}(\epsilon^2),\tag{157}$$

where the $\bullet$ indicates either CE or QE, and

$$M^{\text{CE}}_{\text{gap}} \equiv \frac{6G_3}{\pi^2\ell}\frac{\nu(3+\nu^2)}{(3+5\nu^2)}\Omega^{\text{CE}}_{\text{ext}}, \qquad M^{\text{QE}}_{\text{gap}} \equiv \frac{6G_3\sqrt{1-2H^2}}{\pi^2 L(1-H^2)} = \frac{2}{\Omega^{\text{CE}}_{\text{ext}}} \times M^{\text{CE}}_{\text{gap}}.\tag{158}$$

For the second equality of the quadratic ensemble mass gap we used the relation (25) between $H$ and $\nu$. From (43), very near to extremality we also find

$$T^{\text{QE}} = 2\frac{T^{\text{CE}}}{\Omega^{\text{CE}}_{\text{ext}}} + \mathcal{O}(\epsilon^2).\tag{159}$$

Therefore, in agreement with the ties discussed at the end of Sec. 2.2, the entropies are reporting the same answer: $S^{\text{QE}} = S^{\text{CE}}$ in (157).

A very interesting difference between the ensembles comes in the parameter (scale) that controls the thermodynamic response. Note that the mass gap for the canonical ensemble depends on the angular potential at extremality (45); for the quadratic ensemble this potential is unity at extremality according to (77). This is significant since the expansions in (157) assume large extremal entropy, and therefore from (46) we have

$$S^{\text{CE}}_{\text{ext}} \gg 1 \quad \Rightarrow \quad G_3\Omega^{\text{CE}}_{\text{ext}} \ll 1.\tag{160}$$

That is the angular potential is small in Planck units. This brings some tension to (159): both temperatures differ by big factors making both ensembles fall into different regimes of near-extremality. The canonical ensemble is cooler than the quadratic ensemble.

In the limit $\nu \to 1$ ($H^2 \to 0$) the mass gap of the canonical ensemble diverges, due to $\Omega^{\text{CE}}_{\text{ext}}$. However, in this limit the mass gap for the QE remains finite. Another interesting limit is $\nu \to 0$ ($H^2 = 1/2$); here we find that for both ensembles $M^\bullet_{\text{gap}} = 0$. This is not surprising since when $\nu \to 0$ ($H^2 = 1/2$), we get pure Chern-Simons theory and WBHs are no longer valid solutions.

**Two-point function.** Next we compare the two-point functions (75) and (102). We have

$$\begin{aligned}
G_{\text{CE}}(\omega_{\text{ir}}) &= \left(8\pi\frac{\ell_2^2}{\ell^2}\frac{R_0}{r_0}\frac{\ell}{\beta^{\text{CE}}}\right)^{2\Delta_{\text{CE}}-1} G_{\text{AdS}_2}(\omega_{\text{ir}}),\\
G_{\text{QE}}(\omega_{\text{ir}}) &= \left(\frac{2\pi L}{r_0}\frac{1}{\beta^{\text{QE}}}\right)^{2\Delta_{\text{QE}}-1} G_{\text{AdS}_2}(\omega_{\text{ir}}),
\end{aligned}\tag{161}$$

where, for concreteness, we are defining the AdS$_2$ two point function as

$$G_{\text{AdS}_2}(\omega_{\text{ir}}) = \frac{\Gamma(1-2\Delta_\bullet)\Gamma(\Delta_\bullet)}{\Gamma(2\Delta_\bullet-1)\Gamma(1-\Delta_\bullet)}\frac{\Gamma\left(\Delta_\bullet - i\frac{\ell_2}{\partial}\omega_{\text{ir}}\right)}{\Gamma\left(1-\Delta_\bullet - i\frac{\ell_2}{\partial}\omega_{\text{ir}}\right)},\tag{162}$$

and recall that $\Delta_{\text{CE}} = \Delta_{\text{QE}}$ in the near-extremal limit.

The contrast between the two expressions in (161) is similar to our thermodynamic response. All the scales appearing in $G_{\text{CE}}(\omega_{\text{ir}})$ are roughly order one since $\ell_2 \sim \ell$ and $R_0 \sim r_0$. However, in $G_{\text{QE}}(\omega_{\text{ir}})$ we have that $r_0 \gg L$; recall that $r_0$ controls the extremal entropy in (78) which should be large. If we also take into account the relation (159), we see that $G_{\text{QE}}(\omega_{\text{ir}}) \sim G_{\text{CE}}(\omega_{\text{ir}})$. In other words, for $G_{\text{QE}}(\omega_{\text{ir}})$ to be non-negligible it is natural to scale $T^{\text{QE}}$ with $r_0$.

## 6.2 Comparing perspectives

In this final portion we will take the task of comparing the two-dimensional perspective of Sec. 5, and its own holographic interpretation, against the three-dimensional perspective, and its holographic interpretation in terms of a WCFT.

To start, let us reconcile the semi-classical entropy (157), with the two-dimensional counterpart obtained in Sec. 5.2. From (156), we obtained that the entropy in near-AdS$_2$ is given by

$$S_{2d} = \frac{\pi^2 \ell_2}{6 G_3 \mu} e^{-2\phi_0} \lambda_0 \, T_{2d} \, . \tag{163}$$

Deriving this expression relies on an on-shell analysis of the effective action (123); after renormalizing it, one finds that the entropy comes from the boundary contribution of the Schwarzian action (152). In the following we will write this entropy in terms of the WBH parameters via the dictionary we decoded in (133)-(135). First, it is instructive to relate the ensemble temperatures with $T_{2d}$ in (154); we find

$$e^{-2\phi_0} \lambda_0 T_{2d} = \frac{12 G_3 \mu}{\pi^2 \ell_2} \frac{T^\bullet}{M^\bullet_{\text{gap}}} \, , \tag{164}$$

which holds for both the canonical and quadratic ensemble. In checking this relation one uses in addition, for the canonical ensemble (48) and (50), while for the quadratic ensemble (80) and (82). With this, it is simple to see that

$$S_{2d} = 2 \frac{T^\bullet}{M^\bullet_{\text{gap}}} \, , \tag{165}$$

as expected from the universal behaviour near-extremality of black holes and the expectation that near-AdS$_2$ holography captures this correction correctly. This is another confirmation that the two-dimensional effective action (123) captures correctly the near-extremal regime of warped black holes. At this stage it is also important to remark that our effective theory in AdS$_2$ also captures the leading quantum correction to (165); this follows from the fact that in the fixed angular momentum ensemble the effective action is a Schwarzian term and the results in [38, 86] apply. The quantum entropy is therefore

$$S_{2d} = 2 \frac{T^\bullet}{M^\bullet_{\text{gap}}} + \frac{3}{2} \log T^\bullet + \cdots \, , \tag{166}$$

where the dots are further corrections in $T^\bullet$.

The interesting perspective is to contrast our results on the gravitational side with those from an expected holographic dual. As we reviewed in Sec. 2, there is evidence that WBHs should be interpreted as a warped CFT, either in the canonical or quadratic ensemble, depending on the coordinates used. In particular, we showed that the Wald entropy of non-extremal black holes agrees with the high-temperature behaviour of the partition function of the WCFT; see (20), (37) and discussion within. However, in the present context, we are exploring a near extremal regime, which takes us to low temperatures.

In [29], we derived the near-extremal behaviour of a WCFT, both in the canonical and quadratic ensemble. The key results we obtained are as follows. Adapting to the notation used here, for the canonical ensemble, the near-extremal limit of the WCFT partition function at fixed angular momentum $J$ is

$$Z_J^{\text{CE}}(\beta) = e^{S_0 - \beta E_0} Z_{\text{w-schw.}}(\tilde{\beta}) \, , \tag{167}$$

where

$$Z_{\text{w-schw.}}(\tilde{\beta}) = \left(\frac{\pi}{\tilde{\beta}}\right)^{3/2} \exp\left(\frac{\pi^2}{\tilde{\beta}}\right) \tag{168}$$

is the thermal partition function of the warped Schwarzian sector in a WCFT, and

$$\tilde{\beta} = \frac{3}{c}\sqrt{-\frac{J}{\mathsf{k}}}\,\beta\,. \tag{169}$$

In (167) we also have the contributions for the extremal states, where we have

$$\begin{aligned}
E_0 &= -\sqrt{-\mathsf{k}J} + \dots, \\
S_0 &= 4\pi i P_0^{\text{vac}}\sqrt{-\frac{J}{\mathsf{k}}} + \dots
\end{aligned} \tag{170}$$

The expressions (167)-(170) are valid in the large $c$ limit; consistency of these derivations also requires that $\beta \sim c^\alpha$ and $J \sim c^{2(\alpha-1)}$, with $1 < \alpha \le 3/2$. The dots in (170) are subleading corrections in $J$ and $c$. From (167)-(168) we can read off the leading low-temperature behaviour to be

$$S_{\text{near-wcft}}^{\text{CE}}(\beta) = (1 - \beta\partial_\beta)\ln Z_{\text{w-schw.}} = 2\frac{\pi^2 c}{3}\sqrt{-\frac{\mathsf{k}}{J}}\,T + \frac{3}{2}\log T + \cdots \tag{171}$$

In this expression we are ignoring temperature independent contributions, since they can be viewed as subleading corrections to $S_0$.

The comparison with the gravitational side is excellent. First, the independent parameters here are $\beta$ and $J$, which we naturally match to the gravitional counterpart in Sec. 3.1: $\beta = \beta^{\text{CE}}$ and $J = J_{\text{ext}}^{\text{CE}}$. With this, comparing (170) to the equivalent expressions in (46), we see that $E_0 = M_{\text{ext}}^{\text{CE}}$ and $S^0 = S_{\text{ext}}^{\text{CE}}$ to leading order in the large $c$ limit. The leading temperature response matches: the first term in (171), linear in temperature, agrees with (165) via (50). And the logarithmic correction (171) is exactly what we expect from the quantum corrections from the effective action (152) and (166). All in all, we find perfect agreement between the near-extremal limit of the CE black hole, the near-AdS$_2$ effective description, and the WCFT partition function in the canonical ensemble.

Next, we take the perspective from the WCFT in the quadratic ensemble. The analysis in [29] reports that

$$Z_J^{\text{QE}}(\beta) = e^{S_0 - \beta E_0} Z_{\text{near-QE}}(\tilde{\beta}) \tag{172}$$

is the low temperature partition function at fix $J$ in the quadratic ensemble of a WCFT. Here we have

$$Z_{\text{near-QE}}(\tilde{\beta}) = \left(\frac{\pi}{\tilde{\beta}}\right)^2 \exp\left(\frac{\pi^2}{\tilde{\beta}}\right), \qquad \tilde{\beta} = \frac{12}{c}\beta\,. \tag{173}$$

It is crucial to stress that despite some similarities with (168), there is a different power-law in $\beta$. In (172), the extremal energy and entropy are

$$\begin{aligned}
E_0 &= J + \dots, \\
S_0 &= 2\pi\sqrt{-\langle\mathscr{P}_0\rangle_{\text{vac}}J} + \dots
\end{aligned} \tag{174}$$

Again the dots are subleading corrections in the large $c$ limit, and these expressions are valid when $\beta \sim c^{-\alpha}$ and $J \sim c^{2\alpha}$ for $\alpha > 0$. With this, the low-temperature behaviour of the entropy is

$$S_{\text{near-wcft}}^{\text{QE}}(\beta) = (1 - \beta\partial_\beta)\ln Z_{\text{near-QE}} = 2\frac{\pi^2 c}{12}T + 2\log T + \cdots \tag{175}$$

The comparison with the gravitional side is however problematic. Provided we identify $\beta = \beta^{\text{QE}}$ and $J = -J^{\text{QE}}_{\text{ext}}$, we will find some agreement. More specifically, from (174) and (78), it is straightforward to check that $E_0 = M^{\text{QE}}_{\text{ext}}$ and $S_0 = S^{\text{QE}}_{\text{ext}}$. And it is also simple to check that the linear dependence in entropy in (175) completely agrees with (165) via (82). However, the logarithmic correction in (175) does not match those in the Schwarzian effective action in (152) and (166), and this is a problem. Basically, the near-AdS$_2$ effective theory tells us that the logarithmic corrections in the fixed $(\beta, J)$ ensemble at low temperature should be $3/2 \log T$, and the same correction in the quadratic ensemble of the WCFT does not reproduce this. Although classically the quadratic ensemble seems like a valid choice to setup a holographic dictionary, we are encountering an inconsistency since it is not accounting correctly for the near-extremal entropy. We take this as evidence that at the quantum level the quadractic ensemble does not provide a consistent description of warped black holes.

## 7 Conclusions

We have described several aspects of the near-extremal limit of warped black holes in TMG. Our aim was to contrast any differences or similarities between the canonical and quadratic ensemble. In this context, Sec. 3 and Sec. 4 show compatible results at the classical level, once (41) is taken into account.

One of our main results is to carefully and in full detail construct the near-AdS$_2$ IR effective field theory description of the warped black holes, which contains the JT sector as expected. This is the same theory for both the quadratic and canonical ensemble black hole. The appeal of this theory is that, in addition to accounting correctly for classical aspects, it also accounts for the quantum corrections to the black hole entropy which depend on $\log T$. These corrections can be contrasted with the field theoretic analysis done for WCFT in [29]: only the canonical ensemble of the WCFT reproduces this answer. We find this a useful diagnostic to discriminate between the plethora of ensembles, and asymptotic symmetry groups, that have appeared in the context of three-dimensional black holes.

It is also interesting to comment on how the WAdS/CFT$_2$ proposal stands against this test. In a fixed $J$ and $T$ ensemble we would find agreement between the near-extremal partition function of a CFT$_2$ with (166): both a WCFT and CFT$_2$ report the same answer. The key test here is to analyse the grand canonical ensemble at fixed $\Omega$ and $T$: here is where a WCFT and a CFT$_2$ give a different $\log T$ correction, which is explained in [29]. Along the lines of [59], it would be interesting to work out carefully the boundary conditions in the near-AdS$_2$ region to disentangle carefully what is the near-extremal partition function at fixed $\Omega$ and $T$. In this ensemble we expect the analysis of [64] might be relevant.

Finally, we would like to remark on the unstable mode we have in the near-AdS$_2$ region. This is described around (118)-(122). Our interest here was to highlight the effects of the JT sector on the thermodynamics near extremality, and hence this operator was turned off. It would be interesting to determine if other theories that contain WBHs as solutions also have this unstable mode or not. It is quite possible that this mode is due to instabilities and pathologies of TMG, and absent in other contexts. When the mode is stable in TMG, it corresponds to a relevant operator in the dual theory; it would be interesting to understand what could a WCFT predict about the fate of the system under the presence of a relevant deformation. As argued in [38] there could be instances where it dominates over the JT sector, but the precise effect and at which scale it enters needs to be investigated.

## Acknowledgements

We thank Dio Anninos, Luis Apolo and Monica Guica for useful comments and discussions.

**Funding information** AA is a Research Fellow of the Fonds de la Recherche Scientifique F.R.S.-FNRS (Belgium). AA is partially supported by IISN – Belgium (convention 4.4503.15) and by the Delta ITP consortium, a program of the NWO that is funded by the Dutch Ministry of Education, Culture and Science (OCW). The work of AC has been partially supported by STFC consolidated grant ST/T000694/1. BM is supported in part by the Simons Foundation Grant No. 385602 and the Natural Sciences and Engineering Research Council of Canada (NSERC), funding reference number SAPIN/00047-2020. SD is a Senior Research Associate of the Fonds de la Recherche Scientifique F.R.S.-FNRS (Belgium). SD was supported in part by IISN – Belgium (convention 4.4503.15) and benefited from the support of the Solvay Family. SD acknowledges support of the Fonds de la Recherche Scientifique F.R.S.-FNRS (Belgium) through the CDR project C 60/5 - CDR/OL "Horizon holography: black holes and field theories" (2020-2022), and the PDR/OL C62/5 project "Black hole horizons: away from conformality" (2022-2025).

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
