# Peer review of "Near-Extremal Limits of Warped Black Holes"

_SciPost Physics, doi:SciPost Phys. 15, 083 (2023)_

## Round 1 · Referee Report · Anonymous (Referee 1) · 2023-6-4

Strengths

1-systematically studies and compares the results of warped AdS3 black holes from three perspectives: topologically massive gravity in three dimensions, warped CFTs, and near-AdS2 dynamics
2-finds a way to distinguish the canonical ensemble and quadratic ensemble of WCFT
3-rules out the quadratic ensemble, and hence removes an ambiguity in the holographic duality for warped AdS3 black holes

Report

Warped AdS3 black holes share some features of BTZ black holes and also some features of Kerr black holes in four dimensions. Hence they are useful toy models to describe Kerr black holes holographically. The paper attempts to understand a long-standing ambiguity in the holographic description of warped AdS3 black holes---the choices of different ensembles. To do so, the paper systematically studies warped AdS3 black holes from three perspectives: topologically massive gravity in three dimensions, warped CFTs, and near-AdS2 dynamics.
It is found that the quadratic ensemble is ruled out by considering the logarithmic correction to the entropy. Both the topic and the conclusion are interesting and relevant. The paper is organized and written in a clear and concise way. I recommend the paper to be published in Scipost provided that the authors can make the requested changes listed below. Please note that point-1 is an essential one, and has to be clarified in a satisfactory way .

Requested changes

1- The asymptotic Killing vectors eq. (2.15) in this paper seem to agree with eq. (14) and eq. (15) in arXiv 0808.1911, with N=1, N'=1. If so, the asymptotic algebra eq. (2.17) seems to be different from eq. (17) of arXiv 0808.1911. The authors should clarify this. 2- The paper arXiv 1407.7871 found a set of boundary conditions under which two copies of Virasora algebras were found for warped AdS3. This reference should be added as a piece of evidence of WAdS/CFT2. 3-a typo on page 41: "a expected holographic dual"->"an expected holographic dual"

  • validity: high
  • significance: high
  • originality: high
  • clarity: top
  • formatting: perfect
  • grammar: perfect

Author:  Ankit Aggarwal  on 2023-06-14  [id 3731]

(in reply to Report 2 on 2023-06-04)
Category:
answer to question
reply to objection

We would like to thank the referee for the positive review. We will implement the changes requested in points 2) and 3) in the revised version.

About point 1): The asymptotic Killing vectors in eq. (2.15) of our paper agree with eq. (14) and eq. (15) in arXiv 0808.1911, where one needs to take N=0, N’=1 (and not N=1, N’=1 as stated in the report). Therefore the asymptotic algebra eq. (2.17) agrees with (17) of arXiv 0808.1911. We hope this clarifies this point.

---

## Round 1 · Referee Report · Anonymous (Referee 1) · 2023-6-16

Report

I would like to thank the authors for the reply. All my questions have been answered. I recommend the paper for publication.

---

## Round 1 · Referee Report · Anonymous (Referee 2) · 2023-7-7

Report

In this paper the authors study the holographic implications of having different descriptions --or ensembles-- of a WCFT. It is known that WCFT admit different descriptions, dubbed “canonical” and “quadratic,” related by a state-dependent coordinate transformation. These ensembles give rise to different warped black hole solutions and it is natural to ask how to single out which description is consistent with WAdS/WCFT holography. To answer this question, the authors focus on the near-extremal limit of warped black holes that are solutions of topological massive gravity and the holographic dictionary in their near-AdS2 region.
The authors analyze thermodynamic quantities, correlation functions and construct a low- energy effective theory that describes the near-extremal dynamics of warped black holes and contains a JT sector.

The authors show that at the classical level both descriptions (canonical and quadratic) are equally good. But when looking at quantum corrections to the entropy, only the canonical description agrees with the predictions obtained from field-theoretical analysis of the near-extremal limit of WCFT.

The paper is interesting and adds to the warped black hole literature; I recommend it publication. Below are a couple of suggestions that I think will improve the presentation.

1)To find a solution to the metric perturbation, equation 5.30, the authors turn off
the homogeneous part of the field $\mathcal{F}$. While this is fine to study thermodynamics, $\mathcal{F}_{hom}$ is a massive degree of freedom that could lead to instabilities.
The authors do have a footnote on page 35 about this point. But I think the reader would be better served if this issue is discussed in the body of the paper and not relegated to a footnote.

2)Typos:
page 24: way from extremality -> away from extremality
page 25: grebody -> greybody
page 31: it useful -> it is useful
  • validity: -
  • significance: -
  • originality: -
  • clarity: -
  • formatting: -
  • grammar: -

Author:  Ankit Aggarwal  on 2023-07-10  [id 3793]

(in reply to Report 4 on 2023-07-07)

We would like to thank the referee for the positive remarks and comments. Regarding the two points:

1) We completely agree that this is an important point. We will incorporate footnote 15 in the paragraph below eqn (5.38). We also note that the last paragraph of the conclusion (page 43) also discusses the instability.

2) The typos will be fixed in the resubmission.

---

## Round 2 · Author Response

We hope that with this resubmission we have incorporated the suggestions of referees.

---

## Round 2 · List of Changes

Based on the referee reports: 1. Minor typos corrected. 2. A reference suggested by the referee was added. 3. A footnote was incorporated in the main text.

---

## Editorial Decision

published